



# Comparison of stochastic parameterizations in the framework of a coupled ocean-atmosphere model

Jonathan Demaeyer [1] and Stéphane Vannitsem [1]

[1]Institut Royal Météorologique de Belgique, Avenue Circulaire, 3, 1180 Brussels, Belgium

*Correspondence to:* svn@meteo.be

**Abstract.** A new framework is proposed for the evaluation of stochastic subgrid-scale parameterizations in the context of MAOOAM, a coupled ocean-atmosphere model of intermediate complexity. Two physically-based parameterizations are investigated, the first one based on the singular perturbation of Markov operator, also known as homogenization. The second one is a recently proposed parameterization based on the Ruelle's response theory. The two parameterization are implemented in a
5    rigorous way, assuming however that the unresolved scale relevant statistics are Gaussian. They are extensively tested for a low-order version known to exhibit low-frequency variability, and some preliminary results are obtained for an intermediate-order version. Several different configurations of the resolved-unresolved scale separations are then considered. Both parameterizations show remarkable performances in correcting the impact of model errors, being even able to change the modality of the probability distributions. Their respective limitations are also discussed.


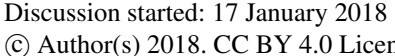

# 1 Introduction

Climate models are not perfect, as they cannot encompass the whole world in their description. Model inaccuracies, also called *model errors*, are therefore always present (Trevisan and Palatella, 2011). One specific type of model error is associated with spatial (or spectral) resolution of the model equations. A stochastic parameterization is a method that allows to represent the

effect of unresolved processes into models. It is a modification, or a closure, of the time-evolution equations for the *resolved variables* that take into account this effect. A typical way to include the impact of these scales is to run a high-resolution models and to perform a statistical analysis to obtain the information needed to compute a closure of the equations such that the truncated model is statistically close to the high-resolution model. These methods are crucial for climate modeling, since the models need to remain as low-resolution as possible, in order to be able to generate runs for very long times. In this case,

the stochastic parameterization should allow for improving the variability and other statistical properties of the climate models at a lower computational cost.

Recently, a revival of interest in stochastic parameterization methods for climate systems has occurred, due to the availability of new mathematical methods to perform the reduction of ordinary differential equations (ODEs) systems. Either based on the conditional averaging (Kifer, 2001; Arnold, 2001; Arnold et al., 2003), on the singular perturbation theory of Markov pro-

cesses (Majda et al., 2001) (MTV), on the conditional Markov chain (Crommelin and Vanden-Eijnden, 2008) or on the Ruelle's response theory (Wouters and Lucarini, 2012) and non-Markovian reduced stochastic equations (Chekroun et al., 2014, 2015). These methods have all in common a rigorous mathematical framework. They provide promising alternatives to other methods such as the ones based on the reinjection of energy from the unresolved scale through backscatter schemes (Frederiksen and Davies (1997); Frederiksen (1999), see also Frederiksen et al. (2017) for a recent review) or on empirical stochastic modeling

methods based on autoregressive processes (Arnold et al., 2013).

The usual way to test the effectiveness of a parameterization method is to consider a well-known climate low-resolution model over which other methods have already been tested. For instance, several methods cited above have been tested on the Lorenz'96 model (Lorenz, 1996), see e.g. Crommelin and Vanden-Eijnden (2008); Arnold et al. (2013); Abramov (2015) and Vissio and Lucarini (2016). These approaches have also been tested in more realistic models of intermediate complexity

that possess a wide range of scales and possibly a lack of timescale separation[1], like for instance the evaluation of the MTV parameterization on barotropic and baroclinic models (Franzke et al., 2005; Franzke and Majda, 2006). Due to the blooming of parameterization methods developed with different statistical or dynamical hypothesis[2], new comparisons are called for.

In this work, we investigate two parameterizations in the context of the MAOOAM ocean-atmosphere coupled model (De Cruz et al., 2016), used as a paradigm for multiscale systems. It is a two-layer baroclinic atmospheric model coupled to a shallow

water ocean. It has been shown to exhibit multiple timescales including a low-frequency variability which is realistic for the midlatitude O-A system (Vannitsem et al., 2015). It possesses also a wide range of behaviors depending on the chosen operating resolution (De Cruz et al., 2016). As such, it forms a nice framework for testing parameterization methods on ocean-atmosphere related problems.

---

[1]Timescale separation, or the existence of a *spectral gap*, is a crucial ingredient over which numerous parameterization methods rely.

[2]i.e. weak coupling hypothesis, large scale separation, ...





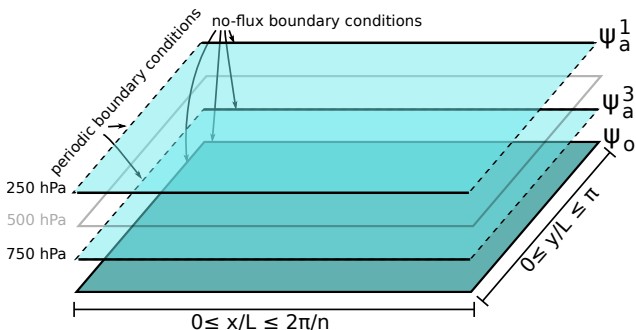

**Figure 1.** MAOOAM model schematic representation.

The particular problem of the atmospheric impact on the ocean could be addressed in this context as in Arnold et al. (2003) and Vannitsem (2014). It is an elegant problem, with a nice timescale separation, and which dates back to the work of Hasselmann (Hasselmann, 1976). However, in the present framework, other arbitrary decompositions of the model are possible, to address other questions. For instance, one may ask the question of the influence of the fast atmospheric processes on the

slower atmospheric mode as well as on the very slow ocean. This problem was already addressed by the authors in Demaeyer and Vannitsem (2017) for a particular decomposition of the atmospheric modes based on the existence of an underlying invariant manifold. The parameterization considered was the one proposed by Wouters and Lucarini (Wouters and Lucarini, 2012), and for this specific invariant-manifold configuration, the stochastic parameterization is greatly simplified. Here, we extend this study by considering also the MTV parameterization (Franzke et al., 2005) with different arbitrary configurations. This in

particular allows us to study different cases with or without multiplicative noise.

This paper is organized as follow: in Section 2, we introduce briefly the MAOOAM model and its time-evolution equations. In Section 3, we review the parameterization methods we have applied to MAOOAM , and detail the model decompositions into resolved and unresolved components. The results obtained with these parameterizations, with different configurations, are presented in Section 4. Finally, the conclusions are provided in Section 5 and new work avenues are proposed.

## 2 The MAOOAM model

The Modular Arbitrary-Order Ocean-Atmosphere Model is a coupled ocean-atmosphere model for midlatitudes. It is composed of a two-layer atmosphere over a shallow-water ocean layer on a $\beta$-plane (De Cruz et al., 2016). The ocean is considered as a closed basin with no-flux boundary conditions, while the atmosphere is defined in a channel, periodic in the zonal direction and with free-slip boundary conditions along the meridional boundaries. The model incorporates both a frictional coupling

(momentum transfer) and an energy balance scheme which accounts for radiative and heat fluxes transfers between the ocean and the atmosphere. This model is developed as a basic representation of the processes at play between the ocean and the atmosphere, and to emphasize their impact on the midlatitude climate and weather. In particular, it has been shown to display a prominent *low-frequency variability* (LFV) in a 36-dimensions model version (Vannitsem et al., 2015).





The dynamical fields of the model include the atmospheric barotropic streamfunction $\psi_\mathrm{a}$ and temperature anomaly $T_\mathrm{a} = 2\frac{f_0}{R}\theta_\mathrm{a}$ (with $f_0$ the Coriolis parameter at midlatitude and $R$ the Earth radius) at 500hPa, as well as the oceanic streamfunction $\psi_\mathrm{o}$ and temperature anomaly $T_\mathrm{o}$. In order to compute the time evolution of these fields, they are expanded in Fourier series with a set of functions satisfying the aforementioned boundary conditions:

$$F_P^A(x',y') = \sqrt{2}\cos(Py') \tag{1}$$

$$F_{M,P}^K(x',y') = 2\cos(Mnx')\sin(Py') \tag{2}$$

$$F_{H,P}^L(x',y') = 2\sin(Hnx')\sin(Py') \tag{3}$$

for the atmosphere and

$$\phi_{H_\mathrm{o},P_\mathrm{o}}(x',y') = 2\sin(\frac{H_\mathrm{o}n}{2}x')\sin(P_\mathrm{o}y') \tag{4}$$

for the ocean, with integer values of $M$, $H$, $P$, $H_\mathrm{o}$, and $P_\mathrm{o}$ and where $x' = x/L$ and $y' = y/L$ are the non-dimensional coordinates on the $\beta$-plane. These functions are then reordered as specified in De Cruz et al. (2016) and the fields are expanded as follow:

$$\psi_\mathrm{a}(x',y',t) = \sum_{i=1}^{n_\mathrm{a}} \psi_{\mathrm{a},i}(t)F_i(x',y'), \tag{5}$$

$$T_\mathrm{a}(x',y',t) = 2\frac{f_0}{R}\sum_{i=1}^{n_\mathrm{a}} \theta_{\mathrm{a},i}(t)\,F_i(x',y'),$$

$$\psi_\mathrm{o}(x',y',t) = \sum_{j=1}^{n_\mathrm{o}} \psi_{\mathrm{o},j}(t)\,(\phi_j(x',y')\,-\,\overline{\phi_j}), \tag{6}$$

$$T_\mathrm{o}(x',y',t) = \sum_{j=1}^{n_\mathrm{o}} \theta_{\mathrm{o},j}(t)\,\phi_j(x',y'). \tag{7}$$

with

$$\overline{\phi_j} = \frac{n}{2\pi^2}\int\limits_0^\pi\int\limits_0^{\frac{2\pi}{n}} \phi_j(x',y')\mathrm{d}x'\mathrm{d}y', \tag{8}$$

a term that allows for mass conservation in the ocean, but does not play any role in the dynamics.

It results into a set of ODEs for the coefficients $\boldsymbol{Z} = (\{\psi_{\mathrm{a},i}\},\{\theta_{\mathrm{a},i}\},\{\psi_{\mathrm{o},j}\},\{\theta_{\mathrm{o},j}\})_{i\in\{1,\dots,n_\mathrm{a}\},j\in\{1,\dots,n_\mathrm{o}\}}$ of the expansion:

$$\dot{Z}_i = H_i + \sum_{j=1}^{N} L_{ij}\,Z_j + \sum_{j,k=1}^{N} B_{ijk}\,Z_j\,Z_k \tag{9}$$

where $N = 2n_\mathrm{a} + 2n_\mathrm{o}$ is the total number of coefficients. This model includes thus forcings, linear interactions and dissipations, as well as quadratic interactions representing the advection terms. Note that the full model equations of MAOOAM include quartic terms for the temperature fields, but those terms have been linearized around equilibrium temperatures (Vannitsem et al., 2015).



## 3  Model decompositions and parameterizations

Now let us consider a more general system of ordinary differential equations (ODEs):

$$\dot{\boldsymbol{Z}} = T(\boldsymbol{Z}) \tag{10}$$

where $\boldsymbol{Z} \in \mathbb{R}^N$ represents the set of variables of a given model. A parameterization of the model supposes first to define a
decomposition of this set of variables into two different subsets $\boldsymbol{Z} = (\boldsymbol{X}, \boldsymbol{Y})$, with $\boldsymbol{X} \in \mathbb{R}^{N_X}$ and $\boldsymbol{Y} \in \mathbb{R}^{N_Y}$. In general, this decomposition is made such that the subsets $\boldsymbol{X}$ and $\boldsymbol{Y}$ have strongly differing *response times*: $\tau_Y \ll \tau_X$ (Arnold et al., 2003). However, we will assume here that this constraint is not necessarily met, allowing for a more arbitrary split of the system variables. System (10) can then be expressed as:

$$\begin{cases} \dot{\boldsymbol{X}} & = & F(\boldsymbol{X}, \boldsymbol{Y}) \\ \dot{\boldsymbol{Y}} & = & H(\boldsymbol{X}, \boldsymbol{Y}) \end{cases} \tag{11}$$

The timescale of the $\boldsymbol{X}$ sub-system is typically (but not always) much longer than the one of the $\boldsymbol{Y}$ sub-system, and it is sometimes materialized by a parameter $\delta = \tau_Y / \tau_X \ll 1$ in front of the time derivative $\dot{\boldsymbol{Y}}$. The $\boldsymbol{X}$ and the $\boldsymbol{Y}$ variables represent respectively the resolved and the unresolved components of the system. A parameterization is a reduction of the system (11) into a closed evolution equation for $\boldsymbol{X}$ alone such that this reduced system approximates the resolved component (Arnold et al., 2003). The term "accurately" here can have several meanings, depending on the kind of problem to solve. For instance we can
ask that the closed system for $\boldsymbol{X}$ has statistics that are very close to the ones of the resolved component of system (11). We can also ask that the trajectories of the closed system remains as close as possible to the trajectories of the full system for long times, but this latter question will not be addressed in the present work.

More precisely a parameterization of the sub-system $\boldsymbol{Y}$ is thus a relation $\Xi$ between the two sub-systems:

$$\boldsymbol{Y} = \Xi(\boldsymbol{X}, t) \tag{12}$$

which allows to effectively close the equations for the sub-system $\boldsymbol{X}$ while retaining the effect of the coupling to the $\boldsymbol{Y}$ sub-system. Since the work of Hasselmann (1976), various stochastic parameterizations have been introduced (Demaeyer and Vannitsem (2018) and Frederiksen et al. (2017) for reviews). They allow for a better representation of the variability of the processes and consider the relation (12) in a statistical sense. In this case, the aforementioned closed system for $\boldsymbol{X}$ becomes a stochastic differential equation (SDE). If the system (11) is rewritten:

$$\begin{cases} \dot{\boldsymbol{X}} & = & F_X(\boldsymbol{X}) + \Psi_X(\boldsymbol{X}, \boldsymbol{Y}) \\ \dot{\boldsymbol{Y}} & = & F_Y(\boldsymbol{Y}) + \Psi_Y(\boldsymbol{X}, \boldsymbol{Y}) \end{cases}, \tag{13}$$

these SDEs are usually written:

$$\dot{\boldsymbol{X}} = F_X(\boldsymbol{X}) + G(\boldsymbol{X}, t) + \boldsymbol{\sigma}(\boldsymbol{X}) \cdot \boldsymbol{\xi}(t) \tag{14}$$

where the matrix $\boldsymbol{\sigma}$, the deterministic function $G$ and the vector of random processes $\boldsymbol{\xi}(t)$ have to be determined. We now detail in the rest of this section the two parameterizations that we will consider, namely the WL and the MTV approach.





### 3.1 Stochastic parameterizations

#### 3.1.1 The Wouters-Lucarini (WL) parameterization

The first one is based on the Ruelle's response theory (Ruelle, 1997, 2009) and is valid when the resolved and the unresolved components are weakly coupled. This method proposed by Wouters and Lucarini (2012) is connected to the Mori-Zwanzig

formalism (Wouters and Lucarini, 2013). It has already been applied to stochastic triads (Wouters et al., 2016; Demaeyer and Vannitsem, 2018), to the Lorenz'96 model (Vissio and Lucarini, 2016) and to the MAOOAM model in the 36-variables configuration displaying LFV (Demaeyer and Vannitsem, 2017). It considers the following decomposition:

$$\dot{\boldsymbol{X}} = F_X(\boldsymbol{X}) + \varepsilon\,\Psi_X(\boldsymbol{X}, \boldsymbol{Y}) \tag{15}$$

$$\dot{\boldsymbol{Y}} = F_Y(\boldsymbol{Y}) + \varepsilon\,\Psi_Y(\boldsymbol{X}, \boldsymbol{Y}) \tag{16}$$

where $\varepsilon$ is a small parameter accounting for the weak coupling and the functions $F_X$ and $F_Y$ account for all the terms involving only $\boldsymbol{X}$ and $\boldsymbol{Y}$, respectively.

As said above, the parameterization is based on the Ruelle's response theory, which quantifies the contribution of the "perturbations" $\Psi_X$ and $\Psi_Y$ to the invariant measure $\rho$ of the fully coupled system (13) as:

$$\rho = \rho_0 + \delta_\Psi \rho^{(1)} + \delta_{\Psi,\Psi} \rho^{(2)} + O(\Psi^3) \tag{17}$$

where $\rho_0$ is the invariant measure of the uncoupled system. The two measures $\rho$ and $\rho_0$ are supposed to be existing, well-defined Sinai-Ruelle-Bower (SRB) measures (Young, 2002). As a result of Eq. (17), the parameterization is based on three different terms having a response similar, up to order two, to the couplings $\Psi_X$ and $\Psi_Y$:

$$\dot{\boldsymbol{X}} = F_X(\boldsymbol{X}) + \varepsilon\,M_1(\boldsymbol{X}) + \varepsilon^2\,M_2(\boldsymbol{X}, t) + \varepsilon^2\,M_3(\boldsymbol{X}, t) \tag{18}$$

where

$$M_1(\boldsymbol{X}) = \left\langle \Psi_X(\boldsymbol{X}, \boldsymbol{Y}) \right\rangle_{\rho_{0,Y}} \tag{19}$$

is an averaging term. $\rho_{0,Y}$ is the measure of the uncoupled system $\dot{\boldsymbol{Y}} = F_Y(\boldsymbol{Y})$. The term $M_2(\boldsymbol{X}, t)$ is a correlation term:

$$\left\langle M_2\big(\boldsymbol{X}(t), t\big) \otimes M_2\big(\boldsymbol{X}(t+s), t+s\big) \right\rangle = \left\langle \Psi'_X(\boldsymbol{X}, \boldsymbol{Y}) \otimes \Psi'_X\big(\phi^s_X(\boldsymbol{X}), \phi^s_Y(\boldsymbol{Y})\big) \right\rangle_{\rho_{0,Y}} \tag{20}$$

where $\otimes$ is the outer product, $\Psi'_X(\boldsymbol{X}, \boldsymbol{Y}) = \Psi_X(\boldsymbol{X}, \boldsymbol{Y}) - M_1(\boldsymbol{X})$ is the centered perturbation and $\phi^s_X$, $\phi^s_Y$ are the flows of the uncoupled system $\dot{\boldsymbol{X}} = F_X(\boldsymbol{X})$ and $\dot{\boldsymbol{Y}} = F_Y(\boldsymbol{Y})$. The $M_3$ term is a memory term:

$$M_3(\boldsymbol{X}, t) = \int\limits_0^\infty \mathrm{d}s\, h(\boldsymbol{X}(t-s), s). \tag{21}$$

involving the memory kernel

$$h(\boldsymbol{X}, s) = \left\langle \Psi_Y(\boldsymbol{X}, \boldsymbol{Y}) \cdot \partial_{\boldsymbol{Y}} \Psi_X\big(\phi^s_X(\boldsymbol{X}), \phi^s_Y(\boldsymbol{Y})\big) \right\rangle_{\rho_{0,Y}} \tag{22}$$



All the averages are thus taken with $\rho_{0,Y}$, and the terms $M_1$, $M_2$ and $M_3$ are derived (Wouters and Lucarini, 2012) such that their responses up to order two match the response of the perturbations $\Psi_X$ and $\Psi_Y$. Consequently, this ensures that for a *weak coupling*, the response of the parameterization (18) on the observables will be approximately the same as the response of the full coupled system.

### 3.1.2 The Majda-Timofeyev-Vanden Eijnde (MTV) parameterization

This approach is based on the singular perturbation methods that were developed for the analysis of the linear Boltzmann equation in an asymptotic limit (Grad, 1969; Ellis and Pinsky, 1975; Papanicolaou, 1976; Majda et al., 2001). These methods are applicable for parameterization purposes if the problem can be cast into a backward Kolmogorov equation. Additionally, the procedure, described in Majda et al. (2001), requires some assumptions on the timescales of the different terms of the system (13). In term of the timescale separation parameter $\delta = \tau_Y/\tau_X$, the fast variability of the unresolved component $Y$ is considered of order $O(\delta^{-2})$, and the coupling terms are acting on an intermediary timescale of order $O(\delta^{-1})$:

$$\dot{X} = F_X(X) + \frac{1}{\delta}\Psi_X(X,Y) \tag{23}$$

$$\dot{Y} = \frac{1}{\delta^2}F_Y(Y) + \frac{1}{\delta}\Psi_Y(X,Y) \tag{24}$$

The intrinsic dynamics $F_X$ is thus the climate $O(1)$ timescale which one is trying to improve with the parameterization. The term $F_Y$ is defined based on which terms are considered as part of the fast variability. It is then assumed that this fast variability can be modeled as an Ornstein-Uhlenbeck process. The Markovian nature of the process defined by Eqs. (23)-(24) and its singular behavior in the limit of an infinite timescale separation ($\delta \to 0$) allow then to apply the method. More specifically, the parameter $\delta$ serves to distinguish terms with different timescales and is then used as a small perturbation parameter. In this setting, the backward Kolmogorov equation reads (Majda et al., 2001):

$$-\frac{\partial \rho^\delta}{\partial s} = \left[\frac{1}{\delta^2}\mathcal{L}_1 + \frac{1}{\delta}\mathcal{L}_2 + \mathcal{L}_3\right]\rho^\delta \tag{25}$$

where the function $\rho^\delta(s, X, Y|t)$ is defined with the final value problem $f(X)$: $\rho^\delta(t, X, Y|t) = f(X)$. The function $\rho^\delta$ can be expanded in term of $\delta$ and inserted in Eq. (25). The zeroth order of this equation $\rho^0$ can be shown to be independent of $Y$ and its evolution given by a closed, averaged backward Kolmogorov equation (Kurtz, 1973):

$$-\frac{\partial \rho^0}{\partial s} = \bar{\mathcal{L}}\rho^0 \tag{26}$$

This equation is obtained in the limit $\delta \to 0$ and gives the sought limiting, averaged process $X(t)$. The parameterization obtained by this procedure is detailed in Franzke et al. (2005) and takes the form (14). As stated above, the parameterization itself depends on which terms of the unresolved component are considered as "fast", and some assumptions should here be made. It is the subject of the next section.





## 3.2 Model decompositions

As MAOOAM is a model whose nonlinearities consist solely of quadratic terms, the decomposition of Eq. (9) into a resolved and an unresolved component can be done based on the constant, linear and quadratic terms of its tendencies, as in Majda et al. (2001) and Franzke et al. (2005) :

$$\dot{\boldsymbol{X}} = \boldsymbol{H}^X + \mathbf{L}^{XX} \cdot \boldsymbol{X} + \mathbf{L}^{XY} \cdot \boldsymbol{Y} + \mathbf{B}^{XXX} : \boldsymbol{X} \otimes \boldsymbol{X} + \mathbf{B}^{XXY} : \boldsymbol{X} \otimes \boldsymbol{Y} + \mathbf{B}^{XYY} : \boldsymbol{Y} \otimes \boldsymbol{Y} \tag{27}$$

$$\dot{\boldsymbol{Y}} = \boldsymbol{H}^Y + \mathbf{L}^{YY} \cdot \boldsymbol{Y} + \mathbf{L}^{YX} \cdot \boldsymbol{X} + \mathbf{B}^{YXX} : \boldsymbol{X} \otimes \boldsymbol{X} + \mathbf{B}^{YXY} : \boldsymbol{X} \otimes \boldsymbol{Y} + \mathbf{B}^{YYY} : \boldsymbol{Y} \otimes \boldsymbol{Y}. \tag{28}$$

The vectors $\boldsymbol{H}$ are the constant terms of the tendencies. The matrices $\mathbf{L}$ give the linear dependencies through matrix-vector products:

$$\left( \mathbf{L}^{XX} \cdot \boldsymbol{X} \right)_i = \sum_{j=1}^{N_X} L^{XX}_{i\,j} \, X_j \,. \tag{29}$$

The symbol $\otimes$ is the outer product and is used to define matrices and tensors, e.g.:

$$(\boldsymbol{X} \otimes \boldsymbol{X})_{ij} = X_i \, X_j \qquad \text{and} \qquad (\boldsymbol{X} \otimes \boldsymbol{X} \otimes \boldsymbol{X})_{ijk} = X_i \, X_j \, X_k \tag{30}$$

and ":" means an element-wise product with summation over the last and first two indices of its first and second arguments respectively[3]. For a rank-3 tensor and a matrix, it gives for example:

$$\left( \mathbf{B}^{XXX} : \boldsymbol{X} \otimes \boldsymbol{X} \right)_i = \sum_{j,k=1}^{N_X} B^{XXX}_{i\,j\,k} \, X_j \, X_k \,. \tag{31}$$

As we have seen in Section 3.1, the two parameterization methods rely on a weak coupling between the components for the WL method and on a clear three-stages timescale separation for the MTV method. Moreover, as we will see below, their decomposition of the equations (27)-(28) is not necessarily the same, depending on modeling choices.

### 3.2.1 Decomposition of the resolved component

For both parameterizations, the decomposition of the $\boldsymbol{X}$ component is the same, and we have:

$$F_X(\boldsymbol{X}) = \boldsymbol{H}^X + \mathbf{L}^{XX} \cdot \boldsymbol{X} + \mathbf{B}^{XXX} : \boldsymbol{X} \otimes \boldsymbol{X} \tag{32}$$

and

$$\Psi_X(\boldsymbol{X}, \boldsymbol{Y}) = \mathbf{L}^{XY} \cdot \boldsymbol{Y} + \mathbf{B}^{XXY} : \boldsymbol{X} \otimes \boldsymbol{Y} + \mathbf{B}^{XYY} : \boldsymbol{Y} \otimes \boldsymbol{Y} \,. \tag{33}$$

---

[3]For two matrices $\mathbf{A}$ and $\mathbf{B}$, it is thus the Frobenius inner product : $\mathbf{A} : \mathbf{B} = \mathrm{Tr}\left( \mathbf{A}^\mathsf{T} \cdot \mathbf{B} \right)$.





### 3.2.2 Decompositions of the unresolved component

The definition of $F_Y$ and $\Psi_Y$ is not necessarily the same for both, but it is of particular importance since it is the measure of the system whose tendencies are given by $F_Y(\boldsymbol{Y})$ over which the averages are performed (Demaeyer and Vannitsem, 2018). Indeed, in the framework of the WL method, it is always the measure of the intrinsic $\boldsymbol{Y}$-dynamics:

$$\dot{\boldsymbol{Y}} = F_Y(\boldsymbol{Y}) = \boldsymbol{H}^Y + \mathbf{L}^{YY} \cdot \boldsymbol{Y} + \mathbf{B}^{YYY} : \boldsymbol{Y} \otimes \boldsymbol{Y} \tag{34}$$

that is used to perform the averaging.

In the framework of the MTV method, the measure of the singular system $\dot{\boldsymbol{Y}} = \frac{1}{\delta^2} F_Y(\boldsymbol{Y})$ is used for the averaging and it is usually assumed that the quadratic $\boldsymbol{Y}$-terms of the unresolved component tendencies represent the fastest timescale of the system (see Majda et al. (2001); Franzke et al. (2005)):

$$F_Y(\boldsymbol{Y}) = \mathbf{B}^{YYY} : \boldsymbol{Y} \otimes \boldsymbol{Y}. \tag{35}$$

and are the ones over which the averaging has to be done. The others terms[4] belong then to $\Psi_Y$. It is interesting to note that there is no a priori justification for this assumption. For instance, the decomposition of the unresolved dynamics could be based on the full intrinsic dynamics (as in the WL method) and $F_Y$ would then be given by the expression (34). We consider these two different assumptions for the MTV in the following.

The MTV and WL parameterizations described above are presented in more details in the appendices A and B, respectively.

### 3.2.3 Noisy model

To take into account model errors or the impact of smaller scales, the present implementation of MAOOAM allows for the addition of Gaussian white noise in each components: resolved and unresolved, for both the ocean and the atmosphere. It also includes the timescale separation parameter $\delta$ of the MTV framework (see Eqs. (23) and (24)) and the coupling parameter $\varepsilon$ of 20   the WL framework (see Eqs. (15) and (16)). The full equation (11) then reads:

$$\dot{\boldsymbol{X}}_{\mathrm{a}} = F_{X,\mathrm{a}}(\boldsymbol{X}) + \boldsymbol{q}_{X,\mathrm{a}} \cdot \mathrm{d}\boldsymbol{W}_{X,\mathrm{a}} + \frac{\varepsilon}{\delta}\,\Psi_{X,\mathrm{a}}(\boldsymbol{X},\boldsymbol{Y}) \tag{36}$$

$$\dot{\boldsymbol{X}}_{\mathrm{o}} = F_{X,\mathrm{o}}(\boldsymbol{X}) + \boldsymbol{q}_{X,\mathrm{o}} \cdot \mathrm{d}\boldsymbol{W}_{X,\mathrm{o}} + \frac{\varepsilon}{\delta}\,\Psi_{X,\mathrm{o}}(\boldsymbol{X},\boldsymbol{Y}) \tag{37}$$

$$\dot{\boldsymbol{Y}}_{\mathrm{a}} = \frac{1}{\delta^2}\left( F_{Y,\mathrm{a}}(\boldsymbol{Y}) + \delta\,\boldsymbol{q}_{Y,\mathrm{a}} \cdot \mathrm{d}\boldsymbol{W}_{Y,\mathrm{a}} \right) + \frac{\varepsilon}{\delta}\,\Psi_{Y,\mathrm{a}}(\boldsymbol{X},\boldsymbol{Y}) \tag{38}$$

$$\dot{\boldsymbol{Y}}_{\mathrm{o}} = \frac{1}{\delta^2}\left( F_{Y,\mathrm{o}}(\boldsymbol{Y}) + \delta\,\boldsymbol{q}_{Y,\mathrm{o}} \cdot \mathrm{d}\boldsymbol{W}_{Y,\mathrm{o}} \right) + \frac{\varepsilon}{\delta}\,\Psi_{Y,\mathrm{o}}(\boldsymbol{X},\boldsymbol{Y}) \tag{39}$$

and the noise amplitude can hence be different for each components. The $\mathrm{d}\boldsymbol{W}$'s vectors are standard Gaussian white noise vectors. In the framework of stochastic parameterization, the presence of noise is sometimes required to smooth the averaging

---

[4]In Franzke et al. (2005), it is also assumed that the constant terms $\boldsymbol{H}^X$ and $\boldsymbol{H}^Y$ are of order $\delta$, making it a four timescales system. These constant terms can thus be neglected in the parameterization. We do not consider this assumption in the present work, and suppose instead that $\boldsymbol{H}^X$ and $\boldsymbol{H}^Y$ are of order one.



measure (Colangeli and Lucarini, 2014) or to increase the small-scale variability to address the problem of "dead" scales. The $F_Y(\boldsymbol{Y}) = (F_{Y,\mathrm{a}}(\boldsymbol{Y}), F_{Y,\mathrm{o}}(\boldsymbol{Y}))$ function can be specified by either Eq. (35) or (34) (only (34) for the WL parameterization). This flexible setup allows for testing both parameterizations in the same framework, with or without additional noise. We now present the results obtained by applying them to the MAOOAM model.

## 4  Results

The relative performance and the interesting features of the parameterizations described in the previous section require to consider multiple versions and resolutions of the model. We thus shall consider in the following two different resolutions. The first one is the 36-variables model version considered in Vannitsem et al. (2015) for which the maximum values for $M$, $H$ and $P$ in Eqs. (1)-(3) is 2. It corresponds to a "$2x$-$2y$" resolution for the atmosphere as referred to in De Cruz et al. (2016). For the ocean, the maximum values for $H_\mathrm{o}$ and $P_\mathrm{o}$ in Eq. (4) are respectively 2 and 4, and the resolution is therefore noted "$2x$-$4y$" in the same notation system. The model version for this first case is thus noted "atm-$2x$-$2y$ oc-$2x$-$4y$" and we shall abbreviate it the acronym "VDDG" from the name of the authors in Vannitsem et al. (2015). The other resolution that we shall consider is "atm-$5x$-$5y$ oc-$5x$-$5y$" that can be abbreviated "$5 \times 5$" since we include Fourier modes up to the wavenumber 5 in both the ocean and the atmosphere. In this latter model version, the model possesses 160 variables.

We shall also consider different sets of parameters for the model configuration. To control the results obtained with the code implementation provided as supplementary material, we will compare our results with those obtained in Demaeyer and Vannitsem (2017) with the first set of parameter defined therein. We will refer to this set of parameter DV2017. A second set of parameters from De Cruz et al. (2016) is also considered and will be denoted DDV2016. For both "DV2017" and "DDV2016", MAOOAM has been shown to depict coupled ocean–atmosphere low-frequency (LFV) variability. In consequence, we will also consider a third parameter set where no LFV is present. The LFV has been removed by reducing by an order of magnitude the wind friction parameter $d$ between the ocean and the atmosphere. The coupled-mode dynamics then disappears. This parameter set is referred as noLFV. In addition, the parameters $\delta$ and $\varepsilon$ appearing in Eqs. (36)-(39) will be set to 1, meaning that we consider the natural timescale separations and coupling strengths of the model. Nevertheless, the study of the impact of these parameters is important (Demaeyer and Vannitsem, 2018) and should be carried out in forthcoming works.

Finally, for a given resolution and a given parameter set, multiple different parameterization experiments can be designed, by using for example the unresolved dynamics (35) or (34) for the MTV parameterization. However, to simplify the study and to be able to compare both the MTV and the WL parameterizations, we will here consider only the dynamics of (34) to perform the averages. Another way of defining different parameterization experiments is by changing the resolved and unresolved components. We shall detail these different experiments and the results obtained in the following subsections.

Unless otherwise specified, the subsequent results were obtained by considering long trajectories lasting $2.8 \times 10^6 \, \mathrm{days} \simeq$ 7680 years, generated with a timestep of 96.9 s, and after an equivalent transient time. Such long trajectories were needed to be able to sample sufficiently the long timescales present in the model ($\simeq$ 20-30 years).





| Parameter | DV2017 | DDV2016 | noLFV |
|---|---|---|---|
| $\lambda$ | 20 | 15.06 | 20 |
| $r$ | $10^{-8}$ | $10^{-7}$ | $10^{-8}$ |
| $d$ | $7.5 \times 10^{-8}$ | $1.1 \times 10^{-7}$ | $10^{-9}$ |
| $C_{\mathrm{o}}$ | 280 | 310 | 350 |
| $C_{\mathrm{a}}$ | 70 | 103.3333 | 100 |
| $k_d$ | $4.128 \times 10^{-6}$ | $2.972 \times 10^{-6}$ | $4.128 \times 10^{-6}$ |
| $k_d'$ | $4.128 \times 10^{-6}$ | $2.972 \times 10^{-6}$ | $4.128 \times 10^{-6}$ |
| $H$ | 500 | 136.5 | 500 |
| $G_{\mathrm{o}}$ | $2.00 \times 10^8$ | $5.46 \times 10^8$ | $2.00 \times 10^8$ |
| $G_{\mathrm{a}}$ | $10^7$ | $10^7$ | $10^7$ |
| $q_{\mathrm{a},X}, q_{\mathrm{a},Y}$ | $5 \times 10^{-4}$ | $5 \times 10^{-4}$ | $5 \times 10^{-4}$ |
| $q_{\mathrm{o},X}, q_{\mathrm{o},Y}$ | 0 | 0 | 0 |

**Table 1.** The main parameters used in the parameterization experiments. For a description of the parameters, see De Cruz et al. (2016) and Demaeyer and Vannitsem (2017).

### 4.1 The 36-variables VDDG model version

For this model version, the atmospheric Fourier modes are denoted:

$$
\begin{aligned}
F_1(x',y') &= \sqrt{2}\cos(y'), \\
F_2(x',y') &= 2\cos(nx')\sin(y'), \\
F_3(x',y') &= 2\sin(nx')\sin(y'), \\
F_4(x',y') &= \sqrt{2}\cos(2y'), \\
F_5(x',y') &= 2\cos(nx')\sin(2y'), \\
F_6(x',y') &= 2\sin(nx')\sin(2y'), \\
F_7(x',y') &= 2\cos(2nx')\sin(y'), \\
F_8(x',y') &= 2\sin(2nx')\sin(y'), \\
F_9(x',y') &= 2\cos(2nx')\sin(2y'), \\
F_{10}(x',y') &= 2\sin(2nx')\sin(2y'),
\end{aligned}
\tag{40}
$$



and the oceanic ones:

$$
\begin{aligned}
\phi_1(x',y') &= 2\sin(nx'/2)\sin(y'), \\
\phi_2(x',y') &= 2\sin(nx'/2)\sin(2y'), \\
\phi_3(x',y') &= 2\sin(nx'/2)\sin(3y'), \\
\phi_4(x',y') &= 2\sin(nx'/2)\sin(4y'), \\
\phi_5(x',y') &= 2\sin(nx')\sin(y'), \\
\phi_6(x',y') &= 2\sin(nx')\sin(2y'), \\
\phi_7(x',y') &= 2\sin(nx')\sin(3y'), \\
\phi_8(x',y') &= 2\sin(nx')\sin(4y').
\end{aligned}
\tag{41}
$$

The associated Fourier coefficients $\{\{\psi_{\mathrm{a},i}\},\{\theta_{\mathrm{a},i}\},\{\psi_{\mathrm{o},j}\},\{\theta_{\mathrm{o},j}\}\}_{i\in\{1,\dots,10\},\,j\in\{1,\dots,8\}}$ form the dynamical system variables. We now propose different partitions of these variables into different resolved and unresolved sets. We will detail, for each of these parameterization experiment, the results obtained for the different aforementioned parameter sets.

### 4.1.1 Parameterization based on the invariant manifold

We first consider a parameterization based on the presence in the VDDG model of an genuine invariant manifold. As stated in Demaeyer and Vannitsem (2017), this manifold is due to the existence of a subset of the Fourier modes that is let invariant by the Jacobian term of the partial differential equation of the system:

$$
J(\psi,\phi) = \frac{\partial\psi}{\partial x}\,\frac{\partial\phi}{\partial y} - \frac{\partial\psi}{\partial y}\,\frac{\partial\phi}{\partial x}.
\tag{42}
$$

In the same spirit, we consider the atmospheric variables outside of this invariant manifold to be unresolved, all the other variables being resolved. The modes $F_2, F_3, F_4, F_7$ and $F_8$ are outside of this invariant set, and therefore the variables

$$\psi_{\mathrm{a},2}, \psi_{\mathrm{a},3}, \psi_{\mathrm{a},4}, \psi_{\mathrm{a},7}, \psi_{\mathrm{a},8}, \theta_{\mathrm{a},2}, \theta_{\mathrm{a},3}, \theta_{\mathrm{a},4}, \theta_{\mathrm{a},7}, \theta_{\mathrm{a},8}$$

are considered as unresolved. Using the same parameterizations, parameters and methods as in Demaeyer and Vannitsem (2017), we should recover the same results with our current implementation[5]. This would thus provide a first mandatory check for the correctness of the current implementation. We show the results obtained on Fig. 2 for the "DV2017" parameter set, where the probability density functions (PDFs) of three important variables of the dynamics are displayed. These three variables are $\psi_{\mathrm{a},1}$, $\psi_{\mathrm{o},2}$ and $\theta_{\mathrm{o},2}$, and were shown in Vannitsem and Ghil (2017) to account for respectively $18\%$, $42\%$ and $51\%$ of the variability in some key reanalysis 2-dimensional fields over the North-Atlantic ocean. In Vannitsem et al. (2015), it was also shown that these three variables are dominant in the LFV pattern found in the VDDG model version of MAOOAM. In addition to the PDFs of the full coupled system (13) and of the uncoupled system $\dot{\boldsymbol{X}} = F_X(\boldsymbol{X})$, the PDFs of the parameterizations are depicted on Fig. 2 with two different settings for the correlation and covariance matrices. Indeed, the two methods need the

---

[5]The code used in Demaeyer and Vannitsem (2017) is different than the implementation provided as supplementary materials with the present work.





specification of the correlation and covariance matrices of the unresolved dynamics (34). These are the matrices $\boldsymbol{\sigma}_Y$, $\boldsymbol{\Sigma}$ and tensor $\boldsymbol{\Sigma}_2$ for the MTV method (see Appendix A), and the matrices $\boldsymbol{\sigma}_Y$, $\langle \boldsymbol{Y} \otimes \boldsymbol{Y}^s \rangle_{\rho_{0,Y}}$ and tensors $\boldsymbol{\rho}_{\partial Y}$, $\boldsymbol{\rho}_{Y \partial Y Y}$ for the WL method (see Appendix B). In the present case, with a parameterization based on the invariant manifold, the dynamics of the unresolved dynamics (34) is a multidimensional Ornstein-Uhlenbeck, for which the measure is well known and thus analytical

expressions for the correlations can be provided to the code. This analytical solution is also used in Demaeyer and Vannitsem (2017) and is one of the setting that we used to compute the parameterization. This setting is also compared with a setting using a numerical least-square regression method to obtain the correlation of variables of the system (34), assuming that these are of the simplified form

$$a \exp\left(-\frac{t}{\tau}\right) \cos(\omega t + k) \tag{43}$$

where $t$ is the time lag and $\tau$ is the decorrelation time. The results obtained for these two different settings of the correlation (analytical vs. numerical) are shown in Fig. 2 with "check" indicating the results obtained with the analytical expressions for the correlations. The latter expressions improve the correction of the dynamics for both the WL and the MTV methods and we recover the same results as in Demaeyer and Vannitsem (2017).[6] On the other hand, in the case where the correlations are specified by the results of the numerical analysis, the parameterizations perform less well, indicating that these methods can be

very sensitive to a correct representation of the correlation structure of the unresolved dynamics.

Both the MTV and the WL methods correct better the oceanic variables whereas the atmospheric variables seem to display too different dynamical behavior between the coupled and uncoupled systems which are difficult to correct. As stated in Demaeyer and Vannitsem (2017), it may be due to the huge dimension of the unresolved system, with half of the atmospheric mode being parameterized. However, the decomposition based on the invariant manifold is an elegant one, based on deep properties of the advection operator (42). It leads to simplified coupling and is thus computationally advantageous. Within this framework,

an adaptation based on the next order of both parameterization methods or the consideration of other parameterizations could lead to a very efficient correction for both the ocean and atmosphere.

As the present implementation allows for an arbitrary selection of the resolved-unresolved components, we shall now consider cases of the VDDG model version with different unresolved components.

### 4.1.2 Parameterization of the wavenumber 2 atmospheric variables

We consider now a smaller set of unresolved variables $\boldsymbol{Y}$, composed of the two higher resolution modes of the model:

$$
\begin{aligned}
F_9(x', y') &= 2\cos(2nx')\sin(2y') \\
F_{10}(x', y') &= 2\sin(2nx')\sin(2y').
\end{aligned}
\tag{44}
$$

The unresolved variables $\boldsymbol{Y}$ are thus the following ones: $\psi_{\mathrm{a},9}$, $\psi_{\mathrm{a},10}$, $\theta_{\mathrm{a},9}$ and $\theta_{\mathrm{a},10}$.

A first comment on the results obtained with that particular configuration is that the WL method seems to be unstable for all

the parameter sets investigated. As stated in the Appendix section B4, the equation (18) is integrated with a Heun algorithm

---

[6]In fact, for the present parameterization based on invariant manifold, the expression of both methods is very close and coincide for a infinite timescale separation.


**Figure 2.** Probability density functions (PDFs) of the dominant variables of the system dynamics with the invariant manifold decomposition for the $\boldsymbol{X}$ and $\boldsymbol{Y}$ components. The densities of the full coupled system (13) and of the uncoupled system are depicted for the "DV2017" parameter set, as well as the ones of the MTV and WL parameterizations and their verifications. The "check" label indicates that the correlations used for the parameterizations were analytical, while for the other they were obtained by numerical analysis.



where the term $M_3$ is considered constant during the corrector-predictor steps. As this could lead to instabilities, a *backward differentiation formulae* (BDF) designed for stiff integro-differential equations has been tested to integrate (18) (Lambert, 1991; Wolkenfelt, 1982), but the instabilities were still present, leading us to suspect a genuine instability in the parameterized system. Indeed, we found that it is the cubic term $\mathbf{M}(s)$ in the memory term $M_3$ (see Eq. (B30)) which causes the divergence,

in particular the interactions with the $F_4$ mode. On the contrary, the MTV parameterization is stable and performs well, despite having a similar cubic term. It indicates that the correlations induced by the memory term are a possible cause of the instability. More research effort to understand this stability issue is needed.

The quality of the solutions obtained with the MTV method alone, applied to the system with the three parameter sets of Table 1, is evaluated using the Kullback-Leibler divergence

$$D_{\mathrm{KL}}(p\|q) = \int\limits_{-\infty}^{\infty} \mathrm{d}x \log \frac{p(x)}{q(x)} \tag{45}$$

between the distribution of $p(x)$ the full coupled system and the distribution $q(x)$ of the parameterized system that is tested. It is related to the information lost when this parameterization is used instead of the full system. The bigger the divergence, the greater is the amount of information lost. For clarity, we show only the divergence of the marginal distributions averaged by component. In every case and for every component, the MTV parameterization reduces dramatically the divergence and thus

corrects well the models, making it clearly a better choice than the "Uncoupled" dynamics. These divergences are compiled in Table 2. It is worth noting that in the atmosphere, it is the barotropic component (the streamfunction) that gets better corrected, while on the contrary, in the ocean, it is the temperature field which benefits the most from the parameterization effects.

As depicted on Figs. 3 and 5, the PDFs of the three dominant variables selected show a neat correction, with a good representation of the LFV when it is present, and a correct shift of the dynamics and the temperature when no LFV occurs. In

particular, one can notice on Fig. 3 a change of the distributions modality due to the parameterization. It raises the question about the mechanism of this rather drastic modification: Is it the noise that changes the dynamics or does the noise simply trigger and shift a bifurcation of the unresolved system, which then induces the change of modality? To give some insight about this latter possibility, we considered the "noLFV" parameter set and increased the ocean-atmosphere wind stress coupling parameter $d$ to $5 \times 10^{-9}$. With this change of parameter, the system now lies nearby the Hopf bifurcation generating the long

periodic orbit which forms the backbone of the LFV. In other words, the system does not exhibit a LFV but a small parameter perturbation could induce it. As seen on Fig. 7, the MTV parameterization then induces a LFV which is not present in both the resolved and the full system. In that case, the parameterization wrongfully "triggers" a bifurcation and thus leads to a false reduced dynamics. This interesting issue, also considered by recent works on the effect of the noise on models dynamics, will be further discussed in the conclusion of this paper (see Section 5).

The impact of the MTV parameterization on the correlation is also particularly important, as seen on Figs. 4 and 6, correcting the covariance (the value at the time lag $t = 0$) and inducing or suppressing oscillations. It shows that the parameterization not only affects the attractors and the climatologies of the models, but also the temporal dynamics.

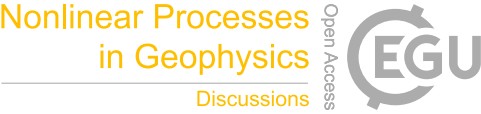



**Figure 3.** Probability density functions (PDFs) of the dominant variables of the system dynamics as in Fig. 2, but for the wavenumber 2 parameterization and with the parameter set "DDV2016". The WL parameterization results are not shown as it is unstable and diverges.

Additional marginal PDF and autocorrelation function figures are available for each variable of the system in the supplementary material. The Kullback-Leibler divergences (45) are available as well.

Since the WL parameterization does not work in the current case, we cannot properly compare both methods. To do so, we shall consider two other parameterization configurations in the following sections.



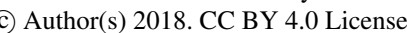


(a) Correlation function of $\psi_{a,1}$.

(b) Correlation function of $\psi_{a,2}$.

(c) Correlation function of $\theta_{a,6}$.

(d) Correlation function of $\theta_{a,8}$.

**Figure 4.** Correlation function of various variables for the wavenumber 2 parameterization and with the parameter set "DDV2016". The correlation of the full coupled system (13), the uncoupled system and the MTV parameterizations are depicted as a function of the time lag $t$. The WL parameterization results are not shown as it is unstable and diverges.

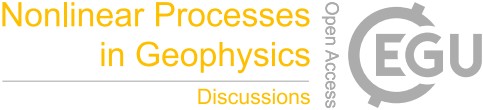



**Figure 5.** Probability density functions (PDFs) of the dominant variables of the system dynamics as in Fig. 2, but for the wavenumber 2 parameterization and with the parameter set "noLFV". The WL parameterization results are not shown as it is unstable and diverges.





(a) Correlation function of $\psi_{a,1}$.

(b) Correlation function of $\theta_{a,2}$.

(c) Correlation function of $\psi_{a,5}$.

(d) Correlation function of $\theta_{a,7}$.

**Figure 6.** Correlation function of various variables for the wavenumber 2 parameterization and with the parameter set "noLFV". The correlation of the full coupled system (13), the uncoupled system and the MTV parameterizations are depicted as a function of the time lag $t$. The WL parameterization results are not shown as it is unstable and diverges.





(a)

(b)

(c)

**Figure 7.** Probability density functions (PDFs) of the dominant variables of the system dynamics as in Fig. 2, but for the wavenumber 2 parameterization. It is shown for the parameter set "noLFV" but with the wind-stress parameter set to $d = 0.5 \times 10^{-9}$ and for which the MTV parameterization depicts a false LFV.





|  | Barotropic atm. | | Baroclinic atm. | | Barotropic oc. | | Temperature oc. | |
|---|---|---|---|---|---|---|---|---|
|  | Uncoupled | MTV | Uncoupled | MTV | Uncoupled | MTV | Uncoupled | MTV |
| DDV2016 | 0.3343 | 0.0075 | 0.1643 | 0.0066 | 0.4592 | 0.0811 | 0.8148 | 0.0346 |
| DV2017 | 0.0776 | 0.0022 | 0.0476 | 0.0031 | 0.1477 | 0.0822 | 0.1886 | 0.0379 |
| noLFV | 0.7415 | 0.0572 | 0.4113 | 0.0386 | 0.0875 | 0.0796 | 1.5815 | 0.2389 |

**Table 2.** Component averaged Kullback-Leibler divergence with respect to the distributions of the full coupled system, in the case of the wavenumber 2 atmospheric variables parameterization.

### 4.1.3 Parameterization of the wavenumber 2 atmospheric baroclinic variables

Let us now consider that only the two baroclinic variables $\theta_{a,9}$ and $\theta_{a,10}$ are unresolved. This particular case allows for the comparison of the WL and the MTV methods. Indeed, in this configuration, the destabilizing cubic interactions with the mode $F_4$ are suppressed and the WL parameterization is now stable. The results are summarized in Table 3 where the Kullback-Leibler divergence (45) of the parameterizations marginal distributions with respect to the full coupled system ones are indicated. The divergence of the uncoupled system with respect to the full system is less pronounced than in the previous section. The uncoupled dynamics is thus not very different from the coupled one. Nevertheless:

– The parameterizations correct quite well the ocean components, except for the MTV parameterization of the baroclinic component in the noLFV case. The MTV parameterization is better in the DDV2016 case, and the WL one is better in the two other cases.

– The barotropic component of the atmosphere is only well corrected in the DV2017 case. Additionally, the MTV fails to correct also the baroclinic PDFs in the two other cases. In fact, the MTV method seems only to performs well in the DV2017 case. Looking to the divergence for every variables (see the supplementary material), we note that those under-performance are due to the incorrect representation of the small-scale wavenumber 2 barotropic variables, namely the $\psi_{a,6}$-$\psi_{a,10}$ ($\psi_{a,5}$-$\psi_{a,10}$) variables for the WL (MTV) method.

Interestingly, the PDFs of the dominant variables $\psi_{a,1}$, $\psi_{o,2}$ and $\theta_{o,2}$ are well corrected as shown in Fig. 8 for the DV2017 case and slightly better corrected in Fig. 10 for the DDV2016 case. We remark that in general the WL parameterization is better at correcting the LFV than the MTV one, but the situation can be more complicated, like in Fig. 10(b), where the WL parameterization captures well one mode of the distribution, and the MTV parameterization captures well another mode.

Regarding the correlation functions, a first general comment is that in this configuration, the decorrelation time of the large scales (mode $F_1$) does not appear to be significantly affected by the absence of the unresolved variables. On the other hand, the impact on the other modes is noticeable, and both parameterizations improve in general the correlations of the resolved variables. Finally, a general observation for all the parameter sets is the bad correction of the variables $\psi_{a,9}$ and $\psi_{a,10}$ (see Fig. 9(d)). This is not surprising since these variables are strongly coupled to the unresolved $\theta_{a,9}$ and $\theta_{a,10}$ variables, and have roughly the same decorrelation timescale. It also may explain the poor scores of the atmospheric zonal wavenumber 2 modes

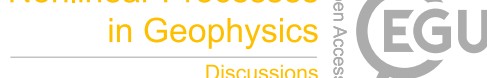


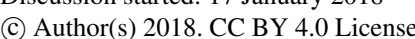

**Figure 8.** Probability density functions (PDFs) of the dominant variables of the system dynamics as in Fig. 2, but for the wavenumber 2 baroclinic parameterization and with the parameter set DV2017.

noted above. It implies that by parameterizing baroclinic variables at certain scales, one should not expect these methods to perform well for the barotropic variables at the same scale.





(a) Correlation function of $\psi_{a,1}$.

(b) Correlation function of $\theta_{a,4}$.

(c) Correlation function of $\theta_{a,7}$.

(d) Correlation function of $\psi_{a,10}$.

**Figure 9.** Correlation function of various variables for the wavenumber 2 parameterization and with the parameter set DV2017. The correlation of the full coupled system (13), the uncoupled system, the MTV and the WL parameterizations are depicted as a function of the time lag $t$.



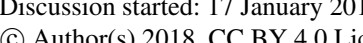



**Figure 10.** Probability density functions (PDFs) of the dominant variables of the system dynamics as in Fig. 2, but for the wavenumber 2 baroclinic parameterization and with the parameter set DDV2016.





|  | Barotropic atm. | | | Baroclinic atm. | | | Barotropic oc. | | | Temperature oc. | | |
|---|---|---|---|---|---|---|---|---|---|---|---|---|
|  | Uncoupled | MTV | WL | Uncoupled | MTV | WL | Uncoupled | MTV | WL | Uncoupled | MTV | WL |
| DDV2016 | 0.0097 | 0.0352 | 0.0104 | 0.0104 | 0.0145 | 0.0032 | 0.1983 | 0.0932 | 0.1120 | 0.1191 | 0.0700 | 0.0930 |
| DV2017 | 0.0110 | 0.0036 | 0.0016 | 0.0018 | 0.0013 | 0.0003 | 0.1312 | 0.0781 | 0.0227 | 0.1277 | 0.0152 | 0.0090 |
| noLFV | 0.0208 | 0.1070 | 0.0433 | 0.0233 | 0.0660 | 0.0174 | 0.1891 | 0.0597 | 0.0523 | 0.1041 | 0.3696 | 0.0998 |

**Table 3.** Component averaged Kuhlback-Leibler divergence with respect to the distributions of the full coupled system, in the case of the wavenumber 2 atmospheric baroclinic variables parameterization.

|  | Barotropic atm. | | | Baroclinic atm. | | | Barotropic oc. | | | Temperature oc. | | |
|---|---|---|---|---|---|---|---|---|---|---|---|---|
|  | Uncoupled | MTV | WL | Uncoupled | MTV | WL | Uncoupled | MTV | WL | Uncoupled | MTV | WL |
| DV2017 | 0.1409 | 0.0376 | 0.0087 | 0.1910 | 0.0252 | 0.0042 | 1.1779 | 0.1254 | 0.2035 | 1.4277 | 0.0613 | 0.1369 |
| noLFV | 0.6905 | 1.1874 | 0.3076 | 0.4360 | 1.0114 | 0.1578 | 0.0214 | 0.1337 | 0.2368 | 1.2016 | 1.1374 | 0.8179 |

**Table 4.** Component averaged Kuhlback-Leibler divergence with respect to the distributions of the full coupled system, in the case of the wavenumber 2 and $F_4$ modes atmospheric parameterization.

### 4.1.4 The parameterization of the atmospheric wavenumber 2 and $F_4$ modes

Another possibility to test the WL parameterization is to remove the aforementioned cubic interactions by considering that the wavenumber 2 modes and the meridional mode

$$F_4(x', y') = \sqrt{2}\cos(2y') \tag{46}$$

are unresolved. In this case the variables, $\psi_{a,4}$, $\theta_{a,4}$, $\psi_{a,9}$, $\psi_{a,10}$, $\theta_{a,9}$ and $\theta_{a,10}$ are considered as unresolved. The WL method is then stable and this configuration allows to compare both parameterizations.

The global averaged Kuhlback-Leibler divergence are given in Table 4 for two parameter sets. These results show a great disparity. For the DV2017 parameter set, the WL method does a better job at correcting the atmosphere while the MTV method corrects better the oceanic modes. Considering the other parameter set without a LFV, we notice that both methods have

troubles at improving the oceanic component. Note that with this particular wind stress reduced configuration, this component dynamics is less important since it can basically be modeled as an atmospheric fluctuations integrator (Hasselmann, 1976). The MTV method has also troubles at modeling the atmospheric dynamics correctly.

The PDFs of the three main variables (see Fig. 11) show that both parameterizations induce a change of modality for $\psi_{a,1}$ and a good correction of the LFV. We note that the MTV method corrects better the LFV signal in the dominant oceanic modes

$\psi_{o,2}$ and $\theta_{o,2}$ (as also shown in Table 4).

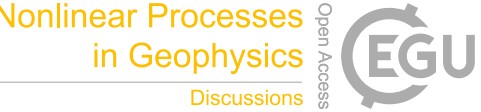



(a)

(b)

(c)

**Figure 11.** Probability density functions (PDFs) of the dominant variables of the system dynamics as in Fig. 2, but for the wavenumber 2 and $F_4$ modes parameterization and with the parameter set "DV2017".



## 4.2 The $5 \times 5$ model version

A higher-resolution test with the MTV method has also been performed by considering the $5 \times 5$ resolution system discussed at section 4. In this version, the resolution goes up to wavenumber 5 in every spatial direction and in every component. We do a quite drastic reduction of the dimensionality of the system by considering every mode above the wavenumber 2 in the

atmosphere as unresolved. Consequently, the uncoupled system is an "atm-$2x$-$2y$ oc-$5x$-$5y$" model version. The result of the parameterization on the dominant $\psi_{o,2}$ and $\theta_{o,2}$ is shown on Fig. 12. The MTV method significantly corrects the 2D marginal distribution compared to the uncoupled model, as seen on the anomaly plots. Therefore, the oceanic section of the attractor of the MTV system is closer to the full coupled system than the uncoupled one. The atmosphere is not very well corrected by the parameterization. This could have multiple cause. First, the trajectory computed at this resolution is shorter (roughly one million

days) and thus some long term equilibration of the dynamics might still take place. In this case, much longer computation might be undertaken. Secondly, the $5 \times 5$ resolution represents a particular case, where the atmosphere's Rhines scale is attained. At this scale, the dynamics is quite different from the higher or lower resolution model version De Cruz et al. (2016). If this has an influence on the parameterization performances, then it calls for even higher-resolution tests for confirmation. Finally, the parameterization may genuinely well correct the ocean while having trouble at improving the atmosphere representation, like

for instance with the invariant manifold parameterization described in section 4.1.1 (see Fig. 2(c)).

## 5 Conclusions

In the present work, we have introduced a new framework to test different stochastic parameterization methods in the context of the ocean-atmosphere coupled model MAOOAM. We have implemented two methods: a homogenization method based on the singular perturbation of Markovian operator and known as MTV (Majda et al., 2001; Franzke et al., 2005), and a method

based on the Ruelle's response theory (Wouters and Lucarini, 2012) abbreviated WL. The code of the program is available as supplementary material, which allows for the future implementation of other parameterization methods in the context of a simplified ocean-atmosphere coupled model.

Within this framework, we have considered two different model resolutions and performed model reduction. We have performed several reductions in the case of a model version with 36 dimensions. We first parameterized atmospheric modes related

to the existence of an invariant manifold present in the dynamics and the results previously obtained in Demaeyer and Vannit-sem (2017) were recovered. In addition, we considered a more complex model reductions by parameterizing the atmospheric wavenumber 2 modes, for different model parameters. In most of the cases, the two methods performed well, correcting the marginal probability distributions and the autocorrelation functions of the model variables, even in the cases where a LFV is developing in the model. However, we have also found that the WL method shows instabilities, due to the cubic interactions

therein. It indicates that the applicability of this method may crucially depend on the long-term correlations in the underlying system. The MTV method does not exhibit this kind of problem.

Additionally, we have found that these methods are able to change correctly the modality of the distributions in some cases. However, in some other cases, they can also trigger a LFV that is absent from the full system. This leads us to underline the





**Figure 12.** Parameterization of the $5 \times 5$ model version with the MTV parameterization: (a) Two-dimensional probability density function (PDF) of the full-coupled dynamics with respect to the two dominant oceanic modes. (b) Anomaly of the PDF of the uncoupled dynamics with respect to the full-coupled one. (c) Anomaly of the PDF of the MTV dynamics with respect to the full-coupled one.





profound impact that a stochastic parameterization, and noise in general, can have on models. For instance, Kwasniok (2014) has shown that the noise can influence the persistence of dynamical regimes and can thus have a non-trivial impact on the PDFs, whose origin is the modification of the dynamical structures by the noise. In the present study as well, the perturbation of the dynamical structures by the noise is a very plausible explanation for the observed change of modality, and for the good
performances of the parameterizations in general. However, if these perturbations can lead to a correct representation of the full dynamics, they can also generate regimes that are not originally present. This may happen near a sensitive bifurcation, as it is the case with a wrong LFV regime, which develops around a long-period periodic orbit (Vannitsem et al., 2015) arising through a Hopf bifurcation. If the main parameter (here for instance the wind stress parameter $d$) is set close to this bifurcation, the noise may thus trigger it while it is not present in the full system (Horsthemke and Lefever, 1984).

The MTV parameterization has also been tested in a intermediate-order version of the model, showing that this parameterization reduces the anomaly of the PDF of the two dominant oceanic modes. The atmospheric modes are however less well corrected. In this case the number of modes that are removed is large and one can wonder whether reducing this number or increasing the resolution will help. More work needs to be done to assess the impact of the parameterizations on higher-order version of the models in the future.

The MTV method is simpler, less involved than the WL one, with no memory term estimations needed, and thus no integrals are being computed at every step. The memory term could however be Markovianized as in Wouters et al. (2016). The interest of the WL method is that it requires only a weak coupling between the resolved and the unresolved components, but no timescale separation between them, which is a desirable feature for a parameterization. Consequently, an improvement of the present framework would be to make the MTV method less dependent on the timescale separation. One way to do that is to
consider the next order in $\delta$, the timescale separation parameter. This can be done effectively using the so-called *Edgeworth* expansion formalism, as shown by Wouters and Gottwald (2017). A next step would thus be to compute this expansion for the present coupled ocean-atmosphere system.

Finally, it would be interesting to consider that the unresolved dynamics used to perform the averaging may have non-Gaussian statistics. In the present work, as stated in Appendix A, the statistics of the neglected variables are assumed to
come from a Gaussian. However, depending on the terms regrouped in this discarded part of the system, the statistics may well be non-Gaussian, and the resulting parameterization developed in this Appendix is then only an approximation. Indeed, unresolved variables with different linear damping terms can lead to such non-Gaussianity (see Sardeshmukh and Penland (2015)). An example of it is the MTV parameterization of the wavenumber 2 in the 36-variable system (see section 4.1.2), where the statistics are weakly non-Gaussian, because the linear damping of the baroclinic unresolved variables is not the same
as the barotropic ones. But other configurations are concerned as well. Taking this into account could probably improve further the results obtained in the present study.



*Code availability.* The source code for MAOOAM v1.2 is available on GitHub at http://github.com/Climdyn/MAOOAM. The stochastic parameterization code is available in fortran in the `stoch` branch of this repository, in the `fortran` subdirectory. It is also provided as supplementary material to the present article.

## Appendix A: The MTV method

We now consider Eqs. (23)-(24) and assume that the dynamics induced by the order $1/\delta^2$ term of the $\boldsymbol{Y}$-dynamics can be replaced by (or behaves like) a multidimensional Ornstein-Uhlenbeck process:

$$\dot{\boldsymbol{Y}} = \frac{1}{\delta^2}\mathbf{A}\cdot\boldsymbol{Y} + \frac{1}{\delta}\mathbf{B}_Y\cdot\mathrm{d}\boldsymbol{W}_Y \tag{A1}$$

where $\boldsymbol{W}_Y$ is a vector of independent Wiener processes. This assumption is described in Majda et al. (2001) as a *working assumption for stochastic modeling*. This assumption is related to other underlying assumptions, i.e. that the dynamics of this

singular term

$$\dot{\boldsymbol{Y}} = \frac{1}{\delta^2}F_Y(\boldsymbol{Y}) \tag{A2}$$

is ergodic and mixing with an integrable decay of correlation (Franzke et al., 2005). We will also assume that this dynamics generates Gaussian distributions, such that the odd-order moments vanish and that the even-order moments are related to the second-order one. In this case, the singular perturbation theory mentioned in section 3.1.2 gives the following result

(see Papanicolaou (1976)) in the limit $\delta \ll 1$ for the parameterization of the resolved component:

$$\dot{\boldsymbol{X}} = F_X(\boldsymbol{X}) + R(\boldsymbol{X}) + G(\boldsymbol{X}) + \sqrt{2}\,\boldsymbol{\sigma}(\boldsymbol{X})\cdot\mathrm{d}\boldsymbol{W} \tag{A3}$$

where

$$R(\boldsymbol{X}) = \frac{1}{\delta}\Big\langle \Psi_X(\boldsymbol{X},\boldsymbol{Y}) \Big\rangle_{\tilde{\rho}_Y} \tag{A4}$$

$$G(\boldsymbol{X}) = \frac{1}{\delta^2}\int_0^\infty \mathrm{d}s \Big\langle \Psi_Y(\boldsymbol{X},\boldsymbol{Y}) \cdot \partial_{\boldsymbol{Y}} \Psi_X(\boldsymbol{X},\boldsymbol{Y}^s) \Big\rangle_{\tilde{\rho}_Y} + \int_0^\infty \mathrm{d}s \Big\langle \Psi_X(\boldsymbol{X},\boldsymbol{Y}) \cdot \partial_{\boldsymbol{X}} \Psi_X(\boldsymbol{X},\boldsymbol{Y}^s) \Big\rangle_{\tilde{\rho}_Y} \tag{A5}$$

and

$$\boldsymbol{\sigma}^2(\boldsymbol{X}) = \mathbf{P}(\boldsymbol{X}) \tag{A6}$$

with

$$\mathbf{P}(\boldsymbol{X}) = \frac{1}{\delta^2}\int_0^\infty \mathrm{d}s \Big\langle \bar{\Psi}_X(\boldsymbol{X},\boldsymbol{Y}) \otimes \bar{\Psi}_X(\boldsymbol{X},\boldsymbol{Y}^s) \Big\rangle_{\tilde{\rho}_Y} \tag{A7}$$

$$\bar{\Psi}_X(\boldsymbol{X},\boldsymbol{Y}) = \Psi_X(\boldsymbol{X},\boldsymbol{Y}) - \Big\langle \Psi_X(\boldsymbol{X},\boldsymbol{Y}) \Big\rangle_{\tilde{\rho}_Y} \tag{A8}$$





and where $\boldsymbol{W}$ is a vector of independent Wiener processes. Note that the terms $G(\boldsymbol{X})$ and $\mathbf{P}(\boldsymbol{X})$ are of order $O(1)$, since the integrals over the lagtime $s$ in Eqs. (A5) and (A7) are of order $O(\delta^2)$. On the other hand, the term $R(\boldsymbol{X})$ is of order $O(1/\delta)$ and identifies with the first term of the left-hand side of Eq. (A25) (see below).

The measure $\tilde{\rho}_Y$ is the one of the system (A2) or the measure of the Ornstein-Uhlenbeck process (A1) replacing it. Similarly,
$\boldsymbol{Y}^s = \phi_Y^s(\boldsymbol{Y})$ is the result of the evolution of the flow $\phi_Y^s$ of the system (A2) or the non-stationary solution of the Ornstein-Uhlenbeck process:

$$\boldsymbol{Y}^s = \exp(\mathbf{A}s/\delta^2) \cdot \boldsymbol{Y} + \frac{1}{\delta} \int_0^s \exp[\mathbf{A}(s-\tau)/\delta^2] \cdot \mathbf{B}_Y \cdot \mathrm{d}\boldsymbol{W}_Y(\tau) \tag{A9}$$

In section A1, we sketch the derivation of Eq. (A3), assuming that the $\boldsymbol{Y}$-dynamics is an Ornstein-Uhlenbeck process. Furthermore, if the $\boldsymbol{Y}$-dynamics is an Ornstein-Uhlenbeck process like (A1), the results (A3)-(A7) can also be expanded to give formula in terms of the covariance and correlation matrices of this process. Therefore, assuming that the dynamics of system (A2) has Gaussian statistics, its measure can be used as $\tilde{\rho}_Y$ and these formula for the process (A1) can then be applied directly using the covariance and correlation matrices of system (A2). It gives a way to practically implement the parameterization as we detail it in section A2.

### A1 Brief sketch of the parameterization derivation

As stated in Section 3.1, the MTV parameterization is based on the singular perturbation theory of Markovian operator. We follow the derivation proposed in Majda et al. (2001) and assume that the singular $O(1/\delta^2)$ term is an Ornstein-Uhlenbeck process like (A1). The backward Kolmogorov equation for Eqs. (23)-(24) for the density $\rho^\delta(s, \boldsymbol{X}, \boldsymbol{Y}|t)$, where $t$ is the final time, reads:

$$-\frac{\partial \rho^\delta}{\partial s} = \left[\frac{1}{\delta^2}\mathcal{L}_1 + \frac{1}{\delta}\mathcal{L}_2 + \mathcal{L}_3\right]\rho^\delta \tag{A10}$$

with the final condition given by some function of $\boldsymbol{X}$:

$$\rho^\delta(t, \boldsymbol{X}, \boldsymbol{Y}|t) = f(\boldsymbol{X}).$$

The operator appearing in Eq. (A10) are given by:

$$\mathcal{L}_1 = \boldsymbol{Y}^\mathsf{T} \cdot \mathbf{A}^\mathsf{T} \cdot \partial_{\boldsymbol{Y}} + \frac{\delta}{2}\mathbf{B}_Y : \partial_{\boldsymbol{Y}} \otimes \partial_{\boldsymbol{Y}} \tag{A11}$$

$$\mathcal{L}_2 = \left(\boldsymbol{Y}^\mathsf{T} \cdot \mathbf{L}^{XX^\mathsf{T}} + \mathbf{B}^{XXY} : \boldsymbol{X} \otimes \boldsymbol{Y} + \mathbf{B}^{XYY} : \boldsymbol{Y} \otimes \boldsymbol{Y}\right) \cdot \partial_{\boldsymbol{X}}$$
$$+ \left(\boldsymbol{X}^\mathsf{T} \cdot \mathbf{L}^{YX^\mathsf{T}} + \boldsymbol{Y}^\mathsf{T} \cdot \mathbf{L}^{YY^\mathsf{T}} + \mathbf{B}^{YXX} : \boldsymbol{X} \otimes \boldsymbol{X} + \mathbf{B}^{YXY} : \boldsymbol{X} \otimes \boldsymbol{Y}\right) \cdot \partial_{\boldsymbol{Y}} \tag{A12}$$

$$\mathcal{L}_3 = \left(\boldsymbol{H}^X + \boldsymbol{X}^\mathsf{T} \cdot \mathbf{L}^{XX^\mathsf{T}} + \mathbf{B}^{XXX} : \boldsymbol{X} \otimes \boldsymbol{X}\right) \cdot \partial_{\boldsymbol{X}} \tag{A13}$$

where $\mathsf{T}$ denotes the matrix transpose operation. Since $\delta \ll 1$, we can now perform an expansion procedure

$$\rho^\delta = \rho_0 + \delta\,\rho_1 + \delta^2\,\rho_2 + \ldots,$$





and recast it in Eq. (A10), equating term by term at each order, to get:

$$\mathcal{L}_1\rho_0 = 0 \tag{A14}$$

$$\mathcal{L}_1\rho_1 = -\mathcal{L}_2\rho_0 \tag{A15}$$

$$\mathcal{L}_1\rho_2 = -\frac{\partial\rho_0}{\partial s} - \mathcal{L}_3\rho_0 - \mathcal{L}_2\rho_1 \tag{A16}$$

5 The solvability condition for theses equations is that the right-hand side belong to the range of $\mathcal{L}_1$ and thus that their average with respect to the invariant measure of (A1). The first solvability conditions is obviously satisfied but the the second is not necessarily satisfied[7] since

$$\mathbb{P}\mathcal{L}_2\rho_0 = \left(\mathbf{B}^{XYY} : \left\langle \boldsymbol{Y} \otimes \boldsymbol{Y} \right\rangle_{\tilde{\rho}_Y}\right) \cdot \partial_{\boldsymbol{X}}\rho_0 = \left(\mathbf{B}^{XYY} : \boldsymbol{\sigma}_Y\right) \cdot \partial_{\boldsymbol{X}}\rho_0$$

where $\mathbb{P}$ is the expectation with respect to the measure of the process (A1) and $\boldsymbol{\sigma}_Y$ is its covariance matrix. It indicates that $1/\delta$

10 effects have to be taken into account in the parameterization. This can be done by following the method to treat "fast-waves" effects described in Majda et al. (2001). From here we thus depart from the standard derivation, to include those $1/\delta$ effects in the parameterization. We need to assume that two timescales are present in the parameterization and consider them in the Kolmogorov equation (A10). Hence, we modify it explicitly by setting:

$$\frac{\partial}{\partial s} \rightarrow \frac{\partial}{\partial s} + \frac{1}{\delta}\frac{\partial}{\partial \tau}$$

15 and

$$\rho^\delta(s, \tau, \boldsymbol{X}, \boldsymbol{Y}|t) = \rho_0(s, \tau, \boldsymbol{X}, \boldsymbol{Y}|t) + \delta\,\rho_1(s, \tau, \boldsymbol{X}, \boldsymbol{Y}|t) + \delta^2\,\rho_2(s, \tau, \boldsymbol{X}, \boldsymbol{Y}|t)\,.$$

Again, recasting this expansion in Eq. (A10) and equating term by term at each order, we obtain:

$$\mathcal{L}_1\rho_0 = 0 \tag{A17}$$

$$\mathcal{L}_1\rho_1 = -\mathcal{L}_2\rho_0 - \frac{\partial\rho_0}{\partial\tau} \tag{A18}$$

20 $$\mathcal{L}_1\rho_2 = -\frac{\partial\rho_0}{\partial s} - \frac{\partial\rho_1}{\partial\tau} - \mathcal{L}_3\rho_0 - \mathcal{L}_2\rho_1 \tag{A19}$$

The solvability condition of Eq. (A17) imposes that $\mathbb{P}\mathcal{L}_1\rho_0 = 0$. Since $\mathbb{P}\mathcal{L}_1\rho_0 = \mathbb{P}\mathcal{L}_1\mathbb{P}\rho_0 = 0$, it means that $\mathbb{P}\rho_0 = \rho_0$ and that $\rho_0$ belongs to the null space of $\mathcal{L}_1$. As a result of the introduction of the $1/\delta$ timescale in the equations, the second solvability condition $\mathbb{P}\mathcal{L}_1\rho_1 = 0$ is now tractable and gives

$$\frac{\partial\rho_0}{\partial\tau} = -\mathbb{P}\mathcal{L}_2\mathbb{P}\rho_0 \tag{A20}$$

25 and thus

$$\rho_1 = -\mathcal{L}_1^{-1}\left(\mathcal{L}_2\rho_0 + \frac{\partial\rho_0}{\partial\tau}\right)\,. \tag{A21}$$

---

[7]The second solvability condition is satisfied if $\mathbf{A}$ in Eq. (A1) is diagonal, as it is the case in Majda et al. (2001) and Franzke et al. (2005). Here we do assume the general case.




These two equations gives together:

$$\rho_1 = -\mathcal{L}_1^{-1}\big(\mathcal{L}_2\rho_0 - \mathbb{P}\mathcal{L}_2\mathbb{P}\rho_0\big). \tag{A22}$$

Since $\mathbb{P}$ commutes with $\mathcal{L}_1^{-1}$ and since $\mathbb{PP} = \mathbb{P}$, we thus have that $\mathbb{P}\rho_1 = 0$ and $\mathbb{P}\partial\rho_1/\partial\tau = 0$. The solvability condition of Eq. (A19) imposes that $\mathbb{P}\mathcal{L}_1\rho_2 = 0$ and thus

$$-\mathbb{P}\frac{\partial\rho_0}{\partial s} = \mathbb{P}\frac{\partial\rho_1}{\partial\tau} + \mathbb{P}\mathcal{L}_3\mathbb{P}\rho_0 + \mathbb{P}\mathcal{L}_2\rho_1 \tag{A23}$$

which becomes, with Eq. (A22),

$$-\frac{\partial\rho_0}{\partial s} = \mathbb{P}\mathcal{L}_3\mathbb{P}\rho_0 - \mathbb{P}\mathcal{L}_2\mathcal{L}_1^{-1}\big(\mathcal{L}_2 - \mathbb{P}\mathcal{L}_2\big)\mathbb{P}\rho_0 \tag{A24}$$

Finally, grouping Eqs. (A20) and (A24) together gives:

$$-\left(\frac{\partial}{\partial s} + \frac{1}{\delta}\frac{\partial}{\partial\tau}\right)\rho_0 = \left[\frac{1}{\delta}\mathbb{P}\mathcal{L}_2\mathbb{P} + \mathbb{P}\mathcal{L}_3\mathbb{P} - \mathbb{P}\mathcal{L}_2\mathcal{L}_1^{-1}\big(\mathcal{L}_2 - \mathbb{P}\mathcal{L}_2\big)\mathbb{P}\right]\rho_0 \tag{A25}$$

which is the result that was obtain in Papanicolaou (1976). From here, the computation proceeds along the standard line described in Majda et al. (2001) and gives the result (A3).

## A2 Practical implementation

We will now derive explicitly the MTV parameterization for the system (27)-(28). For the time of the derivation, we will again assume that the $\boldsymbol{Y}$-dynamics is given by the process (A1), but we suppose that the final results apply as well with the measure,
covariance and correlation matrices of system (A2).

We consider first the case defined by Eq. (35) where the intrinsic dynamics is considered to be given by the quadratic terms of the tendencies alone. We define the matrix

$$\boldsymbol{\Sigma} = \int_0^\infty \mathrm{d}s \left\langle \boldsymbol{Y} \otimes \boldsymbol{Y}^s \right\rangle_{\tilde{\rho}_Y} \tag{A26}$$

and note that

$$\int_0^\infty \mathrm{d}s \left\langle \partial_{\boldsymbol{Y}} \otimes \boldsymbol{Y}^s \right\rangle_{\tilde{\rho}_Y} = \boldsymbol{\sigma}_Y^{-1} \cdot \boldsymbol{\Sigma} \tag{A27}$$

$$\left\langle \boldsymbol{Y} \otimes \partial_{\boldsymbol{Y}} \otimes \boldsymbol{Y}^s \right\rangle_{\tilde{\rho}_Y} = 0 \tag{A28}$$

$$\left\langle \partial_{\boldsymbol{Y}} \otimes \boldsymbol{Y}^s \otimes \boldsymbol{Y}^s \right\rangle_{\tilde{\rho}_Y} = 0 \tag{A29}$$

$$\int_0^\infty \mathrm{d}s \left\langle Y_i \partial_{Y_j} Y_k^s Y_l^s \right\rangle_{\tilde{\rho}_Y} = \int_0^\infty \mathrm{d}s \sum_{m=1}^{N_Y} \sigma_{jm}^{-1}\left(\left\langle Y_m Y_k^s \right\rangle_{\tilde{\rho}_Y}\left\langle Y_i Y_l^s \right\rangle_{\tilde{\rho}_Y} + \left\langle Y_m Y_l^s \right\rangle_{\tilde{\rho}_Y}\left\langle Y_i Y_k^s \right\rangle_{\tilde{\rho}_Y}\right) \tag{A30}$$





since $\left\langle \boldsymbol{Y}^s \right\rangle_{\tilde{\rho}_Y} = 0$ and where $\boldsymbol{\sigma}_Y$ is the covariance matrix of the process (A1). These results can be explicitly obtained using the non-stationary solution (A9). We also define

$$\boldsymbol{\Sigma}_2 = \int\limits_0^\infty \mathrm{d}s \left( \left\langle \boldsymbol{Y} \otimes \boldsymbol{Y}^s \right\rangle_{\tilde{\rho}_Y} \otimes \left\langle \boldsymbol{Y} \otimes \boldsymbol{Y}^s \right\rangle_{\tilde{\rho}_Y} \right) \tag{A31}$$

which is thus a rank-4 tensor. Note that both $\boldsymbol{\Sigma}$ and $\boldsymbol{\Sigma}_2$ are $O(\delta^2)$ objects. With those definitions, it can be shown that the

parameterization (A3) becomes

$$\dot{\boldsymbol{X}} = F_X(\boldsymbol{X}) + R(\boldsymbol{X}) + G(\boldsymbol{X}) + \sqrt{2}\,\boldsymbol{\sigma}^{(1)}(\boldsymbol{X}) \cdot \mathrm{d}\boldsymbol{W}^{(1)} + \sqrt{2}\,\boldsymbol{\sigma}^{(2)} \cdot \mathrm{d}\boldsymbol{W}^{(2)} \tag{A32}$$

with $\left\langle \mathrm{d}\boldsymbol{W}^{(1)}(t) \otimes \mathrm{d}\boldsymbol{W}^{(2)}(t') \right\rangle = 0$ and

$$R(\boldsymbol{X}) = \frac{1}{\delta} \boldsymbol{H}^{(3)} \tag{A33}$$

$$G(\boldsymbol{X}) = \frac{1}{\delta^2} \left[ \sum_{i=0}^2 \boldsymbol{H}^{(i)} + \left( \sum_{i=0}^3 \mathbf{L}^{(i)} \right) \cdot \boldsymbol{X} + \left( \mathbf{B}^{(1)} + \mathbf{B}^{(2)} \right) : \boldsymbol{X} \otimes \boldsymbol{X} + \mathbf{M} :\cdot \boldsymbol{X} \otimes \boldsymbol{X} \otimes \boldsymbol{X} \right] \tag{A34}$$

$$\boldsymbol{\sigma}^{(1)}(\boldsymbol{X}) \cdot \left[ \boldsymbol{\sigma}^{(1)}(\boldsymbol{X}) \right]^\mathsf{T} = \mathbf{P}_1(\boldsymbol{X}) = \frac{1}{\delta^2} \left[ \mathbf{Q}^{(1)} + \mathbf{U} \cdot \boldsymbol{X} + \mathbf{V} : \boldsymbol{X} \otimes \boldsymbol{X} \right] \tag{A35}$$

$$\boldsymbol{\sigma}^{(2)} \cdot \left[ \boldsymbol{\sigma}^{(2)} \right]^\mathsf{T} = \mathbf{P}_2 = \frac{1}{\delta^2} \mathbf{Q}^{(2)} \tag{A36}$$

The product "$:\cdot$" is similar to the definition (31) but with a rank-4 tensor and three variables :

$$\mathbf{M} :\cdot \boldsymbol{X} \otimes \boldsymbol{X} \otimes \boldsymbol{X} = \sum_{j,k,l=1}^{N_X} M_{ijkl}\, X_j\, X_k\, X_l\,. \tag{A37}$$

All the products involved concern the last and the first indices of respectively their first and second arguments. $\mathbf{M}$ and $\mathbf{V}$ are

rank-4 tensors, $\mathbf{U}$ is a rank-3 tensor and the $\mathbf{Q}^{(i)}$'s are matrices. The $\boldsymbol{H}^{(i)}$'s are vectors, the $\mathbf{L}^{(i)}$'s are matrices and the $\mathbf{B}^{(i)}$'s




are rank-3 tensors. The formula of these quantities are given below:

$$H_i^{(0)} = \mathbf{L}_i^{XY} \cdot \left(\boldsymbol{\sigma}_Y^{-1} \cdot \boldsymbol{\Sigma}\right)^\mathsf{T} \cdot \boldsymbol{H}^Y = \sum_{j,k,l=1}^{N_Y} L_{i\,j}^{XY} \Sigma_{jk}^\mathsf{T} \left(\sigma_Y^{-1}\right)_{kl} H_l^Y \tag{A38}$$

$$H_i^{(1)} = \mathbf{B}_i^{XXY} : \left(\mathbf{L}^{XY} \cdot \boldsymbol{\Sigma}\right) = \sum_{j=1}^{N_X} \sum_{k,l=1}^{N_Y} B_{i\,j\,k}^{XXY} L_{j\,l}^{XY} \Sigma_{lk} \tag{A39}$$

$$H_i^{(2)} = \left(\left(\mathbf{B}_i^{XYY} + \mathbf{B}_i^{XYY\,\mathsf{T}}\right) \otimes \mathbf{L}^{YY}\right) \rtimes \left(\boldsymbol{\sigma}_Y^{-1} \cdot \boldsymbol{\Sigma}_2\right) = \sum_{j,k,l,m,n=1}^{N_Y} \left(B_{i\,j\,k}^{XYY} + B_{i\,k\,j}^{XYY}\right) L_{l\,m}^{YY} \left(\sigma_Y^{-1}\right)_{jn} \left(\Sigma_2\right)_{nlkm} \tag{A40}$$

$$5 \quad H_i^{(3)} = \mathbf{B}_i^{XYY} : \boldsymbol{\sigma}_Y = \sum_{j,k=1}^{N_Y} B_{i\,j\,k}^{XYY} \left(\sigma_Y\right)_{jk} \tag{A41}$$

$$L_{ij}^{(0)} = \mathbf{B}_{i\,j}^{XXY} \cdot \left(\boldsymbol{\sigma}_Y^{-1} \cdot \boldsymbol{\Sigma}\right)^\mathsf{T} \cdot \boldsymbol{H}^Y = \sum_{k,l,m=1}^{N_Y} B_{i\,j\,k}^{XXY} \Sigma_{kl}^\mathsf{T} \left(\sigma_Y^{-1}\right)_{lm} H_m^Y \tag{A42}$$

$$L_{ij}^{(1)} = \mathbf{L}_i^{XY} \cdot \left(\boldsymbol{\sigma}_Y^{-1} \cdot \boldsymbol{\Sigma}\right)^\mathsf{T} \cdot \mathbf{L}_j^{YX} = \sum_{k,l,m=1}^{N_Y} L_{i\,k}^{XY} \Sigma_{kl}^\mathsf{T} \left(\sigma_Y^{-1}\right)_{lm} L_{m\,j}^{YX} \tag{A43}$$

$$L_{ij}^{(2)} = \left(\left(\mathbf{B}_i^{XYY} + \mathbf{B}_i^{XYY\,\mathsf{T}}\right) \otimes \mathbf{B}_j^{YXY}\right) \rtimes \left(\boldsymbol{\sigma}_Y^{-1} \cdot \boldsymbol{\Sigma}_2\right) = \sum_{k,l,m,n,p=1}^{N_Y} \left(B_{i\,k\,l}^{XYY} + B_{i\,l\,k}^{XYY}\right) B_{m\,j\,n}^{YXY} \left(\sigma_Y^{-1}\right)_{kp} \left(\Sigma_2\right)_{pmln} \tag{A44}$$

$$L_{ij}^{(3)} = \mathbf{B}_i^{XXY} : \left(\mathbf{B}_j^{XXY} \cdot \boldsymbol{\Sigma}\right) = \sum_{k=1}^{N_X} \sum_{l,m=1}^{N_Y} B_{i\,k\,l}^{XXY} B_{k\,j\,m}^{XXY} \Sigma_{ml} \tag{A45}$$

$$10 \quad B_{ijk}^{(1)} = \mathbf{L}_i^{XY} \cdot \left(\boldsymbol{\sigma}_Y^{-1} \cdot \boldsymbol{\Sigma}\right)^\mathsf{T} \cdot \mathbf{B}_{j\,k}^{YXX} = \sum_{l,m,n=1}^{N_Y} L_{i\,l}^{XY} \Sigma_{lm}^\mathsf{T} \left(\sigma_Y^{-1}\right)_{mn} B_{n\,j\,k}^{YXX} \tag{A46}$$

$$B_{ijk}^{(2)} = \mathbf{B}_{i\,j}^{XXY} \cdot \left(\boldsymbol{\sigma}_Y^{-1} \cdot \boldsymbol{\Sigma}\right)^\mathsf{T} \cdot \mathbf{L}_k^{YX} = \sum_{l,m,n=1}^{N_Y} B_{i\,j\,l}^{XXY} \Sigma_{lm}^\mathsf{T} \left(\sigma_Y^{-1}\right)_{mn} L_{n\,k}^{YX} \tag{A47}$$

$$M_{ijkl} = \mathbf{B}_{i\,j}^{XXY} \cdot \left(\boldsymbol{\sigma}_Y^{-1} \cdot \boldsymbol{\Sigma}\right)^\mathsf{T} \cdot \mathbf{B}_{k\,l}^{YXX} = \sum_{m,n,p=1}^{N_Y} B_{i\,j\,m}^{XXY} \Sigma_{mn}^\mathsf{T} \left(\sigma_Y^{-1}\right)_{np} B_{p\,k\,l}^{YXX} \tag{A48}$$

$$Q_{ij}^{(1)} = \mathbf{L}_i^{XY} \cdot \boldsymbol{\Sigma} \cdot \mathbf{L}_j^{XY\,\mathsf{T}} = \sum_{k,l=1}^{N_Y} L_{i\,k}^{XY} \Sigma_{kl} L_{j\,l}^{XY} \tag{A49}$$

$$Q_{ij}^{(2)} = \left(\mathbf{B}_i^{XYY} \otimes \left(\mathbf{B}_j^{XYY} + \mathbf{B}_j^{XYY\,\mathsf{T}}\right)\right) \rtimes \boldsymbol{\Sigma}_2 = \sum_{k,l,m,n=1}^{N_Y} B_{i\,k\,l}^{XYY} \left(B_{jm\,n}^{XYY} + B_{j\,n\,m}^{XYY}\right) \left(\Sigma_2\right)_{kmln} \tag{A50}$$

$$15 \quad U_{ijk} = \mathbf{L}_i^{XY} \cdot \boldsymbol{\Sigma} \cdot \mathbf{B}_{j\,k}^{XXY} + \mathbf{B}_{i\,j}^{XXY} \cdot \boldsymbol{\Sigma} \cdot \mathbf{L}_k^{XY} = \sum_{l,m=1}^{N_Y} L_{i\,l}^{XY} \Sigma_{lm} B_{j\,k\,m}^{XXY} + \sum_{l,m=1}^{N_Y} B_{i\,j\,l}^{XXY} \Sigma_{lm} L_{k\,m}^{XY} \tag{A51}$$

$$V_{jikl} = \mathbf{B}_{i\,k}^{XXY} \cdot \boldsymbol{\Sigma} \cdot \mathbf{B}_{j\,l}^{XXY} = \sum_{m,n=1}^{N_Y} B_{i\,k\,m}^{XXY} \Sigma_{mn} B_{j\,l\,n}^{XXY} \tag{A52}$$





where · is the product and summation over the last and the first indices of respectively its first and second arguments[8]. The product : is defined as in Eq. (31). The tensors whose indices are indicated define a lower rank tensor, for instance the rank-3 tensor $\mathbf{B}_i^{XXY}$ hence noted defines a matrix whose elements are $B_{i\ j\ k}^{X X Y}$. The symbol $\rtimes$ then defines the following permuted product and summation of two given rank-4 tensors $\mathbf{C}$ and $\mathbf{D}$:

$$\mathbf{C} \rtimes \mathbf{D} = \sum_{ijkl} C_{ijkl} D_{ikjl} \ . \tag{A53}$$

With the notable exception of $\boldsymbol{H}^{(0)}$, $\boldsymbol{H}^{(3)}$ and $\mathbf{L}^{(0)}$, one can easily check that the formulation (A39)-(A52) gives back the formulas of the appendix in Franzke et al. (2005) when $\boldsymbol{\sigma}_Y$ is diagonal. It is these particular tensors that are implemented and computed by the code provided with the present article as supplementary material. They are computed using the covariance matrix $\boldsymbol{\sigma}_Y$ and the integrated correlation matrices $\boldsymbol{\Sigma}$ and $\boldsymbol{\Sigma}_2$ as input. The formulas presented here are valid for an Ornstein-

Uhlenbeck process, but as stated above, the covariance and correlations of the dynamics (A2) can be used directly as well, provided that the right assumptions are fulfilled. In the present work, we have always used the statistic of Eq. (A2) to compute the tensors. Once the tensors are available, the resulting tendencies are then computed at each timestep and Eq. (A3) can be integrated with one of the available integrators.

This solve the case when the singular perturbation $O(1/\delta^2)$ term is given by Eq. (35). In the case where the $O(1/\delta^2)$ term is

given by Eq. (34), i.e. the intrinsic dynamics, it is straightforward to show that the parameterization is exactly the same, except that $\boldsymbol{H}^{(0)} = \boldsymbol{H}^{(2)} = \mathbf{L}^{(0)} = 0$.

## A3 Technical details

The equation (A32) is integrated with a *stochastic Heun algorithm* described in Greiner et al. (1988) and which converges toward the Stratonovitch statistic (Hansen and Penland, 2006). In particular, the $R(\boldsymbol{X})$ and $G(\boldsymbol{X})$ terms and the $\mathbf{P}_1(\boldsymbol{X})$ and $\mathbf{P}_2$

matrix are computed by performing sparse-matrix products. The square root of the matrix $\mathbf{P}_2$ is computed once at initialization with a *singular-value decomposition* (SVD) method to obtain the matrix $\boldsymbol{\sigma}^{(2)}$. The square root of the state-dependent matrix $\mathbf{P}_1(\boldsymbol{X})$ is computed each `mnuti` time[9] to obtain the matrix $\boldsymbol{\sigma}^{(1)}(\boldsymbol{X})$, again with a SVD decomposition. In general, in the present work, we have set `mnuti` equal to the Heun algorithm timestep.

## Appendix B: The WL method

The Wouters-Lucarini method is considered with the decomposition (34) and consists of three terms (Wouters and Lucarini, 2012) :

$$\dot{\boldsymbol{X}} = F_X(\boldsymbol{X}) + \varepsilon M_1(\boldsymbol{X}) + \varepsilon^2 M_2(\boldsymbol{X}, t) + \varepsilon^2 M_3(\boldsymbol{X}, t) \tag{B1}$$

which we detail now in the following.

---

[8]Thus for vectors and matrices it is the standard product.

[9]The notation `mnuti` indicates a parameter of the code implementation, please read the code documentation for more information.





## B1  The $M_1$ term

This is the averaging term, which is defined as

$$M_1(\boldsymbol{X}) = \left\langle \Psi_X(\boldsymbol{X}, \boldsymbol{Y}) \right\rangle_{\rho_{0,Y}} \tag{B2}$$

where $\rho_{0,Y}$ is the measure of the unperturbed intrinsic dynamics (34). Since

$$\Psi_X(\boldsymbol{X}, \boldsymbol{Y}) = \mathbf{L}^{XY} \cdot \boldsymbol{Y} + \mathbf{B}^{XXY} : \boldsymbol{X} \otimes \boldsymbol{Y} + \mathbf{B}^{XYY} : \boldsymbol{Y} \otimes \boldsymbol{Y} \tag{B3}$$

we get the following result:

$$M_1(\boldsymbol{X}) = \mathbf{L}^{XY} \cdot \boldsymbol{\mu}_Y + \mathbf{B}^{XXY} : \boldsymbol{X} \otimes \boldsymbol{\mu}_Y + \mathbf{B}^{XYY} : \boldsymbol{\sigma}_Y \tag{B4}$$

where $\boldsymbol{\mu}_Y = \left\langle \boldsymbol{Y} \right\rangle_{\rho_{0,Y}}$ is the mean of the dynamics (34) and $\boldsymbol{\sigma}_Y$ its covariance matrix. In the following, we shall assume that this dynamics is Gaussian, hence $\boldsymbol{\mu}_Y = 0$ and we get

$$M_1(\boldsymbol{X}) = \mathbf{B}^{XYY} : \boldsymbol{\sigma}_Y \tag{B5}$$

## B2  The $M_2$ term

It is the correlation term, which here can be written as follow

$$M_2(\boldsymbol{X}, t) = \Psi'_{X,1}(\boldsymbol{X})^\mathsf{T} \cdot \mathbf{a}^{XY} \cdot \boldsymbol{\sigma}(t) . \tag{B6}$$

Indeed, since $\Psi'_X(\boldsymbol{X}, \boldsymbol{Y}) = \Psi_X(\boldsymbol{X}, \boldsymbol{Y}) - M_1(\boldsymbol{X})$, $\Psi'_X$ can be decomposed as a product of Schauder basis function of $\boldsymbol{X}$ and $\boldsymbol{Y}$ (Wouters and Lucarini, 2012):

$$\Psi'_X(\boldsymbol{X}, \boldsymbol{Y}) = \sum_{i,j=1}^{2} \Psi'_{X,1,i}(\boldsymbol{X}) \cdot a_{ij}^X \cdot \Psi'_{X,2,j}(\boldsymbol{Y}) \tag{B7}$$

with the basis

$$\Psi'_{X,1,1}(\boldsymbol{X}) = 1 \tag{B8}$$
$$\Psi'_{X,1,2}(\boldsymbol{X}) = \boldsymbol{X}^\mathsf{T} \tag{B9}$$

and

$$\Psi'_{X,2,1}(\boldsymbol{Y}) = \boldsymbol{Y} - \boldsymbol{\mu}_Y \tag{B10}$$
$$\Psi'_{X,2,2}(\boldsymbol{Y}) = vec\left(\boldsymbol{Y} \otimes \boldsymbol{Y} - \boldsymbol{\sigma}_Y\right) \tag{B11}$$

The symbol $vec$ denotes an operation stacking the columns of a matrix into a vector:

$$(vec\,\mathbf{A})_{I(i,j)} = A_{ij} \qquad \text{with} \quad I(i,j) = \text{div}(j,n) + i \tag{B12}$$





where $\mathrm{div}(a,b)$ is the integer division of $a$ by $b$ and $n$ is the second dimension of $\mathbf{A}$. The vector $vec\left(\boldsymbol{Y} \otimes \boldsymbol{Y} - \boldsymbol{\sigma}_Y\right)$ is thus a $N_Y^2$ length vector. The object $a_{ij}^X$ in Eq. (B7) can then be rewritten as a matrix:

$$\mathbf{a}^{XY} = \left[ \begin{array}{cc} \mathbf{L}^{XY} & mat\left(\mathbf{B}^{XYY}\right) \\ \mathbf{B}^{XXY} & 0 \end{array} \right] \tag{B13}$$

where we define

$$\Psi'_{X,1,1}(\boldsymbol{X}) \cdot a_{1j}^X = 1 \cdot a_{1j}^X = a_{1j}^X = \left[ \begin{array}{cc} \mathbf{L}^{XY} & mat\left(\mathbf{B}^{XYY}\right) \end{array} \right] \tag{B14}$$

and

$$\boldsymbol{X}^\mathsf{T} \cdot \mathbf{B}^{XXY} \cdot \boldsymbol{Y} = \mathbf{B}^{XXY} : \boldsymbol{X} \otimes \boldsymbol{Y} \tag{B15}$$

The symbol $mat$ denotes an operation transforming rank-3 tensors into matrices, e.g.:

$$\left(mat\left(\mathbf{B}^{XYY}\right)\right)_{iJ(j,k)} = B_{ijk}^{XYY} \qquad \text{with} \quad J(j,k) = \mathrm{div}(j,n) + k \tag{B16}$$

where $n = N_Y$ is the second dimension of $\mathbf{B}^{XYY}$. With this notation, we have thus for instance:

$$mat\left(\mathbf{B}^{XYY}\right) \cdot vec\left(\boldsymbol{Y} \otimes \boldsymbol{Y}\right) = \mathbf{B}^{XYY} : \boldsymbol{Y} \otimes \boldsymbol{Y} \tag{B17}$$

Here $mat\left(\mathbf{B}^{XYY}\right)$ is thus a $N_X \times N_Y^2$ matrix. The notation $vec$ and $mat$ introduced here are defined such that the expression (B7) can now be written in the compact form with matricial products:

$$\Psi'_X(\boldsymbol{X},\boldsymbol{Y}) = \Psi'_{X,1}(\boldsymbol{X}) \cdot \mathbf{a}_{ij}^X \cdot \Psi'_{X,2}(\boldsymbol{Y}). \tag{B18}$$

Now, the process $\boldsymbol{\sigma}(t)$ appearing in Eq. (B6) has then to be constructed (see Wouters and Lucarini (2012)) such that

$$\left\langle \boldsymbol{\sigma}(t) \otimes \boldsymbol{\sigma}(t+s) \right\rangle = \mathbf{g}(s) \tag{B19}$$

with

$$\mathbf{g}(s) = \left\langle \Psi'_{X,2}(\boldsymbol{Y}) \otimes \Psi'_{X,2}(\boldsymbol{Y}^s) \right\rangle_{\rho_{0,Y}} \tag{B20}$$

$$= \left[ \begin{array}{cc} \left\langle (\boldsymbol{Y} - \boldsymbol{\mu}_Y) \otimes (\boldsymbol{Y}^s - \boldsymbol{\mu}_Y) \right\rangle_{\rho_{0,Y}} & \left\langle (\boldsymbol{Y} - \boldsymbol{\mu}_Y) \otimes vec\left(\boldsymbol{Y}^s \otimes \boldsymbol{Y}^s - \boldsymbol{\sigma}_Y\right) \right\rangle_{\rho_{0,Y}} \\ \left\langle vec\left(\boldsymbol{Y} \otimes \boldsymbol{Y} - \boldsymbol{\sigma}_Y\right) \otimes (\boldsymbol{Y}^s - \boldsymbol{\mu}_Y) \right\rangle_{\rho_{0,Y}} & \left\langle vec\left(\boldsymbol{Y} \otimes \boldsymbol{Y} - \boldsymbol{\sigma}_Y\right) \otimes vec\left(\boldsymbol{Y}^s \otimes \boldsymbol{Y}^s - \boldsymbol{\sigma}_Y\right) \right\rangle_{\rho_{0,Y}} \end{array} \right] \tag{B21}$$

where $\boldsymbol{Y}^s = \phi_Y^s(\boldsymbol{Y})$ is the result of the time-evolution of the flow of the uncoupled $\boldsymbol{Y}$-dynamics given by system (34). Now if we assume that this latter dynamics is Gaussian, the non-diagonal entries in the matrix above vanish, and we can write

$$\boldsymbol{\sigma}(t) = \left[ \begin{array}{c} \boldsymbol{\sigma}_1(t) \\ vec\left(\boldsymbol{\sigma}_2(t)\right) \end{array} \right] \tag{B22}$$




with

$$\left\langle \boldsymbol{\sigma}_1(t) \otimes \boldsymbol{\sigma}_1(t+s) \right\rangle = \left\langle \boldsymbol{Y} \otimes \boldsymbol{Y}^s \right\rangle_{\rho_{0,Y}} - \boldsymbol{\mu}_Y \otimes \boldsymbol{\mu}_Y \tag{B23}$$

$$\left\langle \boldsymbol{\sigma}_1(t) \otimes vec\left(\boldsymbol{\sigma}_2(t+s)\right) \right\rangle = 0 \tag{B24}$$

$$\left\langle vec\left(\boldsymbol{\sigma}_2(t)\right) \otimes vec\left(\boldsymbol{\sigma}_2(t+s)\right) \right\rangle = \left\langle vec\left(\boldsymbol{Y} \otimes \boldsymbol{Y}\right) \otimes vec\left(\boldsymbol{Y}^s \otimes \boldsymbol{Y}^s\right) \right\rangle_{\rho_{0,Y}} - \left\langle vec\left(\boldsymbol{\sigma}_Y\right) \otimes vec\left(\boldsymbol{\sigma}_Y\right) \right\rangle_{\rho_{0,Y}}. \tag{B25}$$

where $\boldsymbol{\sigma}_2$ is a $N_Y \times N_Y$ *matrix* of processes. The $M_2$ term can thus be written:

$$M_2(\boldsymbol{X},t) = \mathbf{L}^{XY} \cdot \boldsymbol{\sigma}_1(t) + \mathbf{B}^{XXY} : \boldsymbol{X} \otimes \boldsymbol{\sigma}_1(t) + \mathbf{B}^{XYY} : \boldsymbol{\sigma}_2(t) \tag{B26}$$

### B3   The $M_3$ term

It is the memory term which is defined as

$$M_3(\boldsymbol{X},t) = \int_0^\infty \mathrm{d}s\, h(\boldsymbol{X}(t-s),s) \tag{B27}$$

with

$$h(\boldsymbol{X},s) = \left\langle \Psi_Y(\boldsymbol{X},\boldsymbol{Y})^\mathsf{T} \cdot \partial_{\boldsymbol{Y}} \Psi_X(\boldsymbol{X}^s,\boldsymbol{Y}^s) \right\rangle_{\rho_{0,Y}} \tag{B28}$$

$$= \left\langle \left( \mathbf{L}^{YX} \cdot \boldsymbol{X} + \mathbf{B}^{YXX} : \boldsymbol{X} \otimes \boldsymbol{X} + \mathbf{B}^{YXY} : \boldsymbol{X} \otimes \boldsymbol{Y} \right)^\mathsf{T} \right.$$
$$\left. \cdot \partial_{\boldsymbol{Y}} \left( \mathbf{L}^{XY} \cdot \boldsymbol{Y} + \mathbf{B}^{XXY} : \boldsymbol{X}^s \otimes \boldsymbol{Y} + \mathbf{B}^{XYY} : \boldsymbol{Y} \otimes \boldsymbol{Y} \right) \right\rangle_{\rho_{0,Y}} \tag{B29}$$

where $\boldsymbol{X}^s = \phi_X^s(\boldsymbol{X})$ is the result of the time-evolution of the flow of the uncoupled $\boldsymbol{X}$-dynamics given by the system $\dot{\boldsymbol{X}} =$

$F_X(\boldsymbol{X})$. Detailing each term in this expression, $M_3$ can be rewritten as

$$M_3(\boldsymbol{X},t) = \int_0^\infty \mathrm{d}s \left[ \left( \mathbf{L}^{(1)}(s) + \mathbf{L}^{(2)}(s) \right) \cdot \tilde{\boldsymbol{X}} + \mathbf{B}^{(1)}(s) : \tilde{\boldsymbol{X}} \otimes \tilde{\boldsymbol{X}} + \mathbf{B}^{(2)}(s) : \tilde{\boldsymbol{X}} \otimes \tilde{\boldsymbol{X}}^s + \mathbf{M}(s) :\cdot\, \tilde{\boldsymbol{X}} \otimes \tilde{\boldsymbol{X}} \otimes \tilde{\boldsymbol{X}}^s \right]_{\tilde{\boldsymbol{X}} = \boldsymbol{X}(t-s)}$$
$$\tag{B30}$$





with $\tilde{X}^s = \phi_X^s\left(X(t-s)\right)$ and

$$L_{ij}^{(1)}(s) = \mathbf{L}_i^{XY} \cdot \boldsymbol{\rho}_{\partial Y}(s)^\mathsf{T} \cdot \mathbf{L}_j^{YX} = \sum_{k,l=1} L_{i\,k}^{XY} \, \rho_{\partial Y}(s)_{kl}^\mathsf{T} \, L_{l\,j}^{YX} \tag{B31}$$

$$L_{ij}^{(2)}(s) = \mathbf{B}_j^{YXY\,\mathsf{T}} : \left(\boldsymbol{\rho}_{Y\partial YY}(s) : \mathbf{B}_i^{XYY}\right) = \sum_{k,l,m,n=1}^{N_Y} B_{l\,j\,k}^{YXY} \, \rho_{Y\partial YY}(s)_{klmn} \, B_{i\,mn}^{XYY} \tag{B32}$$

$$B_{ijk}^{(1)}(s) = \mathbf{L}_i^{XY} \cdot \boldsymbol{\rho}_{\partial Y}(s)^\mathsf{T} \cdot \mathbf{B}_{j\,k}^{YXX} = \sum_{l,m=1}^{N_Y} L_{i\,l}^{XY} \, \rho_{\partial Y}(s)_{ml} \, B_{m\,j\,k}^{YXX} \tag{B33}$$

$$B_{ijk}^{(2)}(s) = \mathbf{B}_{i\,j}^{XXY} \cdot \boldsymbol{\rho}_{\partial Y}(s)^\mathsf{T} \cdot \mathbf{L}_k^{YX} = \sum_{l,m=1}^{N_Y} B_{i\,j\,l}^{XXY} \, \rho_{\partial Y}(s)_{ml} \, L_{mk}^{YX} \tag{B34}$$

$$M_{ijkl}(s) = \mathbf{B}_{i\,j}^{XXY} \cdot \boldsymbol{\rho}_{\partial Y}(s)^\mathsf{T} \cdot \mathbf{B}_{k\,l}^{YXX} = \sum_{m,n=1}^{N_Y} B_{i\,j\,m}^{XXY} \, \rho_{\partial Y}(s)_{nm} \, B_{n\,k\,l}^{YXX} \tag{B35}$$

since

$$\left\langle \boldsymbol{Y} \otimes \partial_{\boldsymbol{Y}} \otimes \boldsymbol{Y}^s \right\rangle_{\rho_{0,Y}} = 0 \tag{B36}$$

$$\left\langle \partial_{\boldsymbol{Y}} \otimes \boldsymbol{Y}^s \otimes \boldsymbol{Y}^s \right\rangle_{\rho_{0,Y}} = 0 \tag{B37}$$

because the dynamics of (34) is assumed to be Gaussian and where

$$\boldsymbol{\rho}_{\partial Y} = \left\langle \partial_{\boldsymbol{Y}} \otimes \boldsymbol{Y}^s \right\rangle_{\rho_{0,Y}} = \boldsymbol{\sigma}_Y^{-1} \cdot \left\langle \boldsymbol{Y} \otimes \boldsymbol{Y}^s \right\rangle_{\rho_{0,Y}} \tag{B38}$$

$$\boldsymbol{\rho}_{Y\partial YY} = \left\langle \boldsymbol{Y} \otimes \partial_{\boldsymbol{Y}} \otimes \boldsymbol{Y}^s \otimes \boldsymbol{Y}^s \right\rangle_{\rho_{0,Y}} \tag{B39}$$

and the expression of this last definition can be inferred from Eq. (A30).

## B4 Technical details

The tendencies appearing in Eq. (B1) are computed according to the same Heun algorithm described in Sec. A3. A small difference with the latter here is that the term $M_3$, due to its computational cost, is computed every `muti` time[10] and remains constant in between. The term $M_1$ is simply obtained by computing sparse-matrix products. The process $\boldsymbol{\sigma}(t)$ of the term $M_2$ is emulated by a Multidimensional $n$-th order Auto-Regressive process (MAR($n$)) described in Penny and Harrison (2007). The parameters and the order $n$ of this process can be assessed for instance by using the variational Bayesian approach described in this article. The integrand of the term $M_3$ is obtained by computing sparse-matrix products and integrating for each timestep the uncoupled $\boldsymbol{X}$-dynamics forward, to obtain $\tilde{\boldsymbol{X}}^s$. The integral is then performed with a simple trapezoidal rule with a timestep `muti` over a time interval which is set by the parameter `meml`. This interval should be set so that the absolute value of the integrand has sufficiently decreased at the end of it. In practice, it is sufficient to set `meml` proportional to the longest decorrelation time of the uncoupled $\boldsymbol{Y}$-dynamics.

---

[10]The notation `muti` indicates a parameter of the code implementation, please read the code documentation for more information.



*Competing interests.* Stéphane Vannitsem is a member of the editorial board of the journal. Jonathan Demaeyer declares that he has no conflict of interest.

*Acknowledgements.* The authors thank Lesley De Cruz for her authorization to use the Figure 1. This work is supported by the Belgian Federal Science Policy Office under contracts BR/121/A2/STOCHCLIM.





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
