# Peer review of "Comparison of stochastic parameterizations in the framework of a coupled ocean-atmosphere model"

_Nonlinear Processes in Geophysics, 2017_

## Referee Comment (RC1) · Anonymous Referee #1 · 5 Feb 2018

In the present paper the authors present an in-depth comparison of two methods of parameterisation. The methods compared are homogenisation as used by Majda et al. and the weak coupling method as developed by Wouters and Lucarini. The model used is the MAOOAM spectral coupled ocean- atmosphere model. Both the model and the methods used are relevant. The present work adds a considerably to the existing evidence of the efficacity of stochastic methods of parameterisation.

The model is used not very realistic, but it is at the upper boundary of the size of system that can currently be intensively studied numerically, so I consider it to be a good choice. The chosen parameterization methods are to my knowledge the only ones with a derivation from first principle, making them sound choices. Perhaps a comparison to a data-driven method would have further improved the work, but I don't consider this to be a necessary requirement.

Another drawback of this study is the assumption that $\varepsilon = \delta = 1$. It is encouraging that even with this rather crude assumption the methods show a good result, but it does make one wonder whether the shortfalls of some of the methods are innate to the method or are due to the assumptions made. Calculating the decorrelation time scales of the slow and fast dynamics and setting $\delta = \tau_Y / \tau_X$ would be a less crude assumption.

The metrics of comparison are appropriate and to my opinion constitute sufficient evidence for the conclusions the authors have drawn.

Based on the manuscript and the research scope of the journal, I believe this work falls within the desired scope of NPG and meets the review criteria.

I have a few minor suggestions that I list below as bullet points.

- p.2 l.22: over which → on which

- p.2 footnote: over which → on which

- p.3 l.5: mode → modes

- p.4 l.20: "It results": what does "it" refer to? Please clarify.

- p.6 eq.17: there is an $\varepsilon$ and $\varepsilon^2$ missing in the second and third term on the RHS

- p.7 l.5: Vanden-Eijnden

- p.7 l.10: In term → In terms

- p.7 eq.23-24: note that in homogenization theory $F_X$ and $F_Y$ can also depend on both $X$ and $Y$

- p.8 eq.32-33: The choice of decomposition is arbitrary without information of the size of different terms in the dynamical equation. If you agree, please insert a comment (here or elsewhere) that other decompositions may improve or decrease the performance. Could you also comment why you consider the assumptions of a time-scale separation or weak coupling to be valid in the current system?

- p.9 l.4-10: Either method should work as long as the fast dynamics has an ergodic invariant measure (with some extra assumptions). Please comment on the reason for choosing two different decompositions for the two methods.

- p.9 l.13-14: "we consider these two different assumptions": this is contradictory to p.10 l.27: "we will here consider only the dynamics of (34)".

- p.10 l.1: please add a few words explaining the problem of "dead" scales

- p.13 l.6: setting $\rightarrow$ settings

- p.30 l.12: "We will also assume...": this is not an extra assumption if you already consider a Gaussian Ornstein-Uhlenbeck process.

- p.30 eq. A5 and A7: the $1/\delta^2$ factors here should be 1

- p.31 l.17: the backward KE describes the evolution of expectation values, not densities

- p.32 l.5-6: incomplete sentence

- p.32 footnote: Additional to the diagonality, one needs assumption A4 of Majda et al. (2001)

- p.34 eqs A34: check the factors $1/\delta^2$

- p.36 l. 11-12: Have any tests been performed to conclude that the process can be approximated with an Ornstein-Uhlenbeck?

---

## Referee Comment (RC2) · Anonymous Referee #2 · 19 Apr 2018

Review of "Comparison of stochastic parameterizations in the framework of a coupled ocean-atmosphere model" by Demaeyer and Vannitsem

Recommendation: Minor revisions

1) The correlation plots (Figs. 4, 6 and 9) do not seem to be autocorrelations but autocovariances since the value at lag 0 is not 1. I think it would be easier to compare if the authors plot the autocorrelation function.

2) The are a few recent review papers on stochastic modeling. Including them in the introduction would make it more informative: Berner, Judith, et al. "Stochastic parameterization: Toward a new view of weather and climate models." Bulletin of the American

[Figure]

Meteorological Society 98.3 (2017): 565-588.

Franzke, Christian LE, et al. "Stochastic climate theory and modeling." Wiley Interdisciplinary Reviews: Climate Change 6.1 (2015): 63-78.

3) In the introduction the authors seem to distinguish between stochastic parameterizations and backscatter schemes. Backscatter schemes can also be stochastic so can be similar. I also think stochastic parameterizations are implicitly also based on the idea to "backscatter" energy from the unresolved scales into the resolved.

4) Page 2, footnote 2: I do not understand the meaning here.

5) MAOOAM should be defined at first use.

6) Are "weak coupling" and "time scale separation" equivalent in a mathematical sense? How would one measure weak coupling in the real atmos-ocean system? For me weak coupling is rather opaque concept whereas time-scale separation is more tangible (at least I know how to estimate this from real data). Some comments on this would be appreciated.

7) In the "seamless MTV" procedure of Franzke et al. (2005) we generalized MTV and do not need to assume an Ornstein-Uhlenbeck process any longer but just one stochastic process with Gaussian statistics.

8) In Eqs. 36-39 are the coupling and time-scale separation parameters included. Which values have been used for the experiments?

9) In my work I use a split integration scheme: Runge-Kutta 4th order for the deterministic part and Euler-Maruyama for the stochastic part. Such a split scheme might solve parts of your numerical problems.

10) In MTV the cubic terms are nonlinear damping. This has been shown in Majda, Andrew J., Christian Franzke, and Daan Crommelin. "Normal forms for reduced stochastic climate models." Proceedings of the National Academy of Sciences 106.10

(2009): 3649-3653.

Peavoy, Daniel, Christian LE Franzke, and Gareth O. Roberts. "Systematic physics constrained parameter estimation of stochastic differential equations." Computational Statistics & Data Analysis 83 (2015): 182-199.

Perhaps one can also ensure that the cubic term in WL is negative definite then the system should be stable.

11) The manuscript would benefit from a careful proofreading.

---

## Referee Comment (RC3) · Anonymous Referee #3 · 8 May 2018

This very nice work presents an in depth numerical analysis of two stochastic parametrization schemes, homogenization (aka MTV) and the recently proposed method by Wouters and Lucarini (WL), and their ability to be used to model unresolved scales in an underresolved model. The work uses a coupled ocean-atmosphere model of intermediate complexity, MAOOOAM, and the authors have made their code publicly available. Independent of the actual valuable results which the authors report on, the simple fact that the paper introduces this software package makes this manuscript in my view extremely valuable and will have some positive impact. This is very commendable.

[Figure]

The authors find that both parametrization methods are able to capture the empirical probability function of the full system. Their performance was found to be very sensitive to a good estimate of the correlation structure of the unresolved scales.

I have only some minor comments:

1) At the beginning of Section 3.1.2 the authors state that "These methods are applicable for parameterization purposes if the problem can be cast into a backward Kolmogorov equation." This is not a correct statement: Every dynamical system (deterministic or stochastic) can be cast into a backward Kolmogorov equation. Casting a nonlinear dynamical system into the linear backward Kolmogorov equation (or alternatively the linear Fokker-Planck equation) allows for the machinery of perturbation theory of linear systems to derive explicit formulae for the homogenized equations. The authors may want to reference here the monograph "Multiscale Methods: Averaging and Homogenization" by Pavliotis and Stuart.

Furthermore, the authors provide references for the mathematical justification of MTV (including Papanicolaou). These are references if the underlying dynamical system is stochastic; for the applicability of the MTV procedure for deterministic systems, the references are Melbourne and Stuart (2011), Nonlinearity 24, pp 1361, Gottwald and Melbourne (2013), Proc. Roy. Soc. A 469, pp 2013020 and Kelly and Melbourne (2017), Journal of Functional Analysis 10, pp 4063.

The choice of notation with \rho for the backward Kolmogorov equation which propagates expectation values, is odd. \rho is usually reserved for densities. I also found the backward in time integration with the expectation value being defined at the final time cumbersome; why not have a positive sign on the left hand side of (25) and (26), and have the expectation value equal to $f(X)$ at t=0? Also, the generator \mathcal{L} is not defined when it is introduced.

2) In the case when the unresolved scales consist of the wavenumber 2 atmospheric variables, the authors found that the WL approach leads to unstable dynamics. I am

surprised by that. The authors relate this to a cubic form. Can the authors comment on which assumption of WL is being violated by this choice of unresolved variables? Are higher-order corrections needed?

There are a few typos in the text:

Eqn (24): missing full stop.

Above (42): let -> left

---

## Author Comment (AC1) · 18 Jun 2018

**Comparison of stochastic parameterizations in the framework of a coupled ocean-atmosphere model – Response to the 1st reviewer**

Jonathan Demaeyer [1] and Stéphane Vannitsem [1]

[1]Institut Royal Météorologique de Belgique, Avenue Circulaire, 3, 1180 Brussels, Belgium

**1 General comments**

We thank the reviewer for his/her carefull reading of the manuscript. We will first adress the general comments and then address the proposed suggestions.

First, a comparison with a data-driven method would have been very interesting too. It could be easily done in the framework
of the software code given as supplementary material. However, we wanted that the article remained focused on *first principle* methods and their technical details.

Secondly, the reviewer indicates that the consideration of the case $\varepsilon = \delta = 1$ only is another drawback of the study. We wanted to focus on the methods derivation within the ocean-atmosphere model, and their performance in its original form. The extension to other values has been done in Demaeyer and Vannitsem (2018) with a simpler system. It can of course be
reproduced here but it is left for future works.

**2 Modifications**

We now address the proposed suggestions:

- p.2 l.22: over which → on which

  **Answer:** Ok.
* * *
- p.2 footnote: over which → on which

  **Answer:** Ok.
* * *
- p.3 l.5: mode → modes

  **Answer:** Ok.

– p.4 l.20: "It results": what does "it" refer to? Please clarify.

  **Answer:** Indeed, the procedure was not correctly explained. We have modified the text according to:

> *By recasting these expansions into the partial differential equations of the model, one obtains a set of ODEs for its coefficients $\boldsymbol{Z} = (\{\psi_{a,i}\}, \{\theta_{a,i}\}, \{\psi_{o,j}\}, \{\theta_{o,j}\})_{i \in \{1,\dots,n_a\}, j \in \{1,\dots,n_o\}}$ :*

– p.6 eq.17: there is an $\varepsilon$ and $\varepsilon^2$ missing in the second and third term on the RHS

  **Answer:** Ok.

– p.7 l.5: Vanden-Eijnden

  **Answer:** Ok.

– p.7 l.10: In term → In terms

  **Answer:** Ok.

– p.7 eq.23-24: note that in homogenization theory $F_X$ and $F_Y$ can also depend on both $\boldsymbol{X}$ and $\boldsymbol{Y}$

  **Answer:** We have added this comment as footnote in the text:

> *Note that in homogenization theory $F_X$ and $F_Y$ can also depend on both $\boldsymbol{X}$ and $\boldsymbol{Y}$, a possibility that was not considered here in order to effectively compare the MTV and WL parameterizations.*

– p.8 eq.32-33: The choice of decomposition is arbitrary without information of the size of different terms in the dynamical equation. If you agree, please insert a comment (here or elsewhere) that other decompositions may improve or decrease the performance. Could you also comment why you consider the assumptions of a time-scale separation or weak coupling to be valid in the current system?

  **Answer:** We have change this section according to:

*This decomposition can be chosen arbitrarily since the only requirement is that $F_X$ depends solely on $\boldsymbol{X}$. However, in the following (and in the provided code), we will consider that the resolved-unresolved components form a coupled system, with "maximal" uncoupled dynamics. In this view for both parameterizations, the decomposition of the $\boldsymbol{X}$ component is the same:*

$$F_X(\boldsymbol{X}) = \boldsymbol{H}^X + \mathbf{L}^{XX} \cdot \boldsymbol{X} + \mathbf{B}^{XXX} : \boldsymbol{X} \otimes \boldsymbol{X}$$

*and*

$$\Psi_X(\boldsymbol{X}, \boldsymbol{Y}) = \mathbf{L}^{XY} \cdot \boldsymbol{Y} + \mathbf{B}^{XXY} : \boldsymbol{X} \otimes \boldsymbol{Y} + \mathbf{B}^{XYY} : \boldsymbol{Y} \otimes \boldsymbol{Y} \,.$$

*This choice was also consider in Franzke et al. (2005), dealing with the MTV parameterization. It is worth noting that other decompositions may improve or decrease the performance of the parameterization.*

In addition, we do not assume that the resolved-unresolved components decomposition possesses a weak coupling or a timescale separation since these can be set arbitrarily by the user of the code. If the decomposition is for instance a resolved ocean and an unresolved atmosphere, then a timescale separation exists (weak-coupling is less obvious due to the temperature scheme). On the other hand, for all the cases considered in the result section, the decomposition is made in the atmosphere, where no weak-coupling or timescale separation (spectral gap) is found.

So the assumptions are valid depending on the "experiment" which is made. We have developed the parameterizations *based* on these assumptions, but then one can "push" these methods over their limits, in order to see if they would satisfy more difficult applications, like the parameterization of some atmospheric large scales. This is what we have tried to do.
* * *
– p.9 l.4-10: Either method should work as long as the fast dynamics has an ergodic invariant measure (with some extra assumptions). Please comment on the reason for choosing two different decompositions for the two methods.

**Answer:** We have modified this section according to:

*The definition of $F_Y$ and $\Psi_Y$ is also arbitrary, but it is of particular importance since it is the measure of the system whose tendencies are given by $F_Y(\boldsymbol{Y})$ over which the averages are performed (De-maeyer and Vannitsem, 2018). A requirement is thus that the dynamics $\dot{\boldsymbol{Y}} = F_Y(\boldsymbol{Y})$ has an ergodic invariant measure. In the framework of the WL method, it is natural to consider the measure of the intrinsic $\boldsymbol{Y}$-dynamics as:*

$$F_Y(\boldsymbol{Y}) = \boldsymbol{H}^Y + \mathbf{L}^{YY} \cdot \boldsymbol{Y} + \mathbf{B}^{YYY} : \boldsymbol{Y} \otimes \boldsymbol{Y}$$

*to perform the averaging.*

*In the framework of the MTV method, the measure of the $O(1/\delta^2)$ singular system $\dot{\boldsymbol{Y}} = F_Y(\boldsymbol{Y})$ is used for the averaging and it is usually assumed that the quadratic $\boldsymbol{Y}$-terms of the unresolved component tendencies represent the fastest timescale of the system (see Majda et al. (2001); Franzke et al. (2005)):*

$$F_Y(\boldsymbol{Y}) = \mathbf{B}^{YYY} : \boldsymbol{Y} \otimes \boldsymbol{Y}.$$

*and those are the ones over which the averaging has to be done.*

*In both cases, the others terms belong then to $\Psi_Y$.*

*It is interesting to note that there is no a priori justification for one or the other assumption. For both parameterization methods, the decomposition of the unresolved dynamics could be based on Eq. (34) or on Eq. (35). The code provided as supplementary materials allows to select the $F_Y$-dynamics as either Eq. (34) or Eq. (35).*

*The MTV and WL parameterizations described above are presented in more details in the appendices A and B, respectively.*

We have chosen the dynamics of Eq. (34) to test the case with maximal uncoupled resolved-unresolved components. The test with Eq. (35) can easily be done with the code provided. In addition, it has already been used in Franzke et al. (2005).

Following the reviewer comment, we have also noticed that the line 2 p. 10 :

*The $F_Y(\boldsymbol{Y}) = (F_{Y,a}(\boldsymbol{Y}), F_{Y,o}(\boldsymbol{Y}))$ function can be specified by either Eq. (35) or (34) (only (34) for the WL parameterization).*

should be modified. The text inside the parenthesis should be removed since the dynamics (35) could also be used to perform the WL parameterization. The code should allow this dynamics to be used by this parameterization, it will be changed in a future version.

– p.9 l.13-14: "we consider these two different assumptions": this is contradictory to p.10 l.27: "we will here consider only the dynamics of (34)".

**Answer:** Indeed, there is a contradiction. It has been solved by removing the sentence "we consider these two different assumptions..." (p.9 l.13-14).
* * *
– p.10 l.1: please add a few words explaining the problem of "dead" scales

**Answer:** We have decided to remove the expression *"dead" scales* which we have in fact not found in the literature. In place we have written :

[...] *or to increase the small-scale variability to address the problem of scales that are passively slaved and whose variability comes uniquely from their interactions with others.*
* * *
– p.13 l.6: setting → settings

**Answer:** Ok.
* * *
– p.30 l.12: "We will also assume...": this is not an extra assumption if you already consider a Gaussian Ornstein-Uhlenbeck process.

**Answer:** Indeed, we have thus modified and moved this sentence just after equation (A1):

*This dynamics thus generates Gaussian distributions, such that the odd-order moments vanish and that the even-order moments are related to the second-order one.*
* * *
– p.30 eq. A5 and A7: the $1/\delta^2$ factors here should be 1

**Answer:** These factors in front of the time integrals are correct. This is due to the fact that we use the $1/\delta^2$ scaled dynamics Eq. (A2) to compute the average. Therefore, as stated p.31 line 1-2, the integrals over the lagtime $s$ is of order $\delta^2$ and the r.h.s. of Eq. (A5) and (A7) are of order 1. The integrals are of order $\delta^2$ since the exponential decorrelation of the scaled dynamics (A2) are scaled accordingly. In addition, the integrated correlation matrices $\Sigma$ and $\Sigma_2$ are also $O(\delta^2)$ matrices, as stated p. 34 line 4.

In principle, the unscaled dynamics $\dot{Y} = F_Y(Y)$ corresponding to the $\mathcal{L}_1$ operator should be used to compute the average. But here, the need for a code that allows to compute both the MTV and WL parameterization, with the

same initialization files, led to this particular formulation. The text of the appendix try to stay close to how the code performs the computation of the parameterization.

To clarify this for the reader, we added the following footnote p.31 on line 2 :

*It is due to the fact that we use directly the measure $\tilde{\rho}_Y$ of the $O(1/\delta^2)$ dynamics (A2) to performs the averaging, and not the measure of the dynamics $\dot{Y} = F_Y(Y)$.*
* * *
– p.31 l.17: the backward KE describes the evolution of expectation values, not densities

**Answer:** To our knowledge, the backward Kolmogorov equation:

$$-\frac{\partial}{\partial t}p(x,t) = b(x,t)\frac{\partial}{\partial x}p(x,t) + \frac{1}{2}\Sigma(x,t)\frac{\partial}{\partial x^2}p(x,t)$$

describes the evolution of the probability density $p(x,t)$ with a final condition $p(x,s) = u_s(x)$ with $t < s$. But it is true that the Feynman-Kac formula allows for its interpretation in terms of the expectation of the final condition $u_s(x)$ over all paths starting from $x$ at time $t$, see for instance Pavliotis (2014). In that sense, the probability density of a SDE is always an expectation over paths, the transition probability giving the "weight" (measure) of each path. Therefore both formulations are correct and we stick with the word "density" but we add "probability" in front for the sake of clarity. See also Gardiner (2009) and Risken (1996).
* * *
– p.3p.32 l.5-6: incomplete sentence

**Answer:** Has been completed by adding the word "vanishes" at the end.
* * *
– p.32 footnote: Additional to the diagonality, one needs assumption A4 of Majda et al. (2001)

**Answer:** A diagonal matrix $\mathbf{A}$ in Eq. (A1) implies that assumptions A4 is fulfilled, due to the following property of the quadratic terms in quasi-geostrophic systems:

$$\frac{\partial}{\partial Y_i}B^{YYY}_{i\ j\ k}\,Y_j Y_k = 0 \qquad \forall j,k \quad . \tag{1}$$

Indeed, if $\mathbf{A}$ is diagonal then the multidimensional Ornstein-Uhlenbeck process is made of uncorrelated one-dimensional Ornstein-Uhlenbeck processes, and thus:

$$\left\langle B^{YYY}_{i\ j\ k}\,Y_j Y_k \right\rangle = 0 \qquad \forall i,j,k \tag{2}$$

which is precisely the assumption A4. In other words, in this case the covariance matrix of the multidimensional O-U process is also diagonal and with the special property (1), it implies that the assumption A4 is fulfilled.

To explain this better in the article, we removed the footnote and modified the text according to:

> *The first solvability conditions is obviously satisfied but the the second one is not necessarily satisfied*
> *since*
>
> $$\mathbb{P}\mathcal{L}_2\rho_0 = \left(\mathbf{B}^{XYY} : \left\langle \boldsymbol{Y} \otimes \boldsymbol{Y}\right\rangle_{\tilde{\rho}_Y}\right)\cdot\partial_{\boldsymbol{X}}\rho_0 = \left(\mathbf{B}^{XYY} : \boldsymbol{\sigma}_Y\right)\cdot\partial_{\boldsymbol{X}}\rho_0$$
>
> *where $\mathbb{P}$ is the expectation with respect to the measure of the process (A1) and $\boldsymbol{\sigma}_Y$ is its covariance*
> *matrix. If the matrix $\mathbf{A}$ in Eq. (A1) is considered to be diagonal, as in Majda et al. (2001) and*
> *in Franzke et al. (2005), then it is satisfied. Indeed, in this case $\boldsymbol{\sigma}_Y$ is diagonal and $\mathbb{P}\mathcal{L}_2\rho_0 = 0$ due to*
> *the following property of the quadratic terms in the model:*
>
> $$\frac{\partial}{\partial Y_i}B^{YYY}_{i\;j\;k}\,Y_j\,Y_k = 0 \quad .$$
>
> *However, here we do consider the general case: $\mathbf{A}$ is not diagonal and thus we can have $\mathbb{P}\mathcal{L}_2\rho_0 \neq 0$.*
> *It then indicates that $1/\delta$ effects have to be taken into account in the parameterization.*

5      – p.34 eqs A34: check the factors $1/\delta^2$

**Answer:** As stated above, the "integrated" correlation matrices $\boldsymbol{\Sigma}$ and $\boldsymbol{\Sigma}^2$ are of order $\delta^2$, therefore these factors are correct.

     – p.36 l. 11-12: Have any tests been performed to conclude that the process can be approximated with an Ornstein-
10      Uhlenbeck?

**Answer:** No other tests than a rapid evaluation of the PDFs of the processes and a check of the fast exponential decorrelation properties have been done. In general, the PDFs looked Gaussian. Since then, we have computed the standardized third and fourth moments which show that indeed the statistics of the unresolved processes

$$\dot{\boldsymbol{Y}} = F_Y(\boldsymbol{Y})$$

15      is nearly Gaussian for every case that we considered. We comment now on that at the end of the introduction of section 4:

> *In each case, we have also checked the statistics of the dynamics (34) and have concluded that it is*
> *nearly Gaussian.*

However, as stated in the conclusion, the parameterization methods could be extended to deal with the non-Gaussian case.

20    We thank again the reviewer for his/her suggestions.

**References**

Demaeyer, J. and Vannitsem, S.: Stochastic Parameterization of Subgrid-Scale Processes: A Review of Recent Physically Based Approaches, in: Advances in Nonlinear Geosciences, pp. 55–85, Springer, 2018.

Franzke, C., Majda, A. J., and Vanden-Eijnden, E.: Low-order stochastic mode reduction for a realistic barotropic model climate, Journal of the atmospheric sciences, 62, 1722–1745, 2005.

Gardiner, C. W.: Handbook of stochastic methods, Springer Berlin, fourth edn., 2009.

Majda, A. J., Timofeyev, I., and Vanden Eijnden, E.: A mathematical framework for stochastic climate models, Communications on Pure and Applied Mathematics, 54, 891–974, 2001.

Pavliotis, G. A.: Stochastic processes and applications: Diffusion processes, the Fokker-Planck and Langevin equations, vol. 60, Springer, 2014.

Risken, H.: Fokker-planck equation, in: The Fokker-Planck Equation, pp. 63–95, Springer, 1996.

---

## Author Comment (AC2) · 18 Jun 2018

**Comparison of stochastic parameterizations in the framework of a coupled ocean-atmosphere model – Response to the 2nd reviewer**

Jonathan Demaeyer [1] and Stéphane Vannitsem [1]

[1]Institut Royal Météorologique de Belgique, Avenue Circulaire, 3, 1180 Brussels, Belgium

We thank the reviewer for his/her comments on the manuscript and address his/her proposed suggestions:

1. The correlation plots (Figs. 4, 6 and 9) do not seem to be autocorrelations but autocovariances since the value at lag 0 is not 1. I think it would be easier to compare if the authors plot the autocorrelation function.

   **Answer:** Depending on the background, "autocovariance" can mean "autocorrelation". In this case, what the reviewer

5  denotes as "autocorrelation" would be called "normalized autocorrelation". Besides these terminology issues, we believe that the "autocovariances" are more complete because they they illustrate both the time scale and variability properties of the variable.
* * *
2. The are a few recent review papers on stochastic modeling. Including them in the introduction would make it more

10  informative: Berner, Judith, et al. "Stochastic parame- terization: Toward a new view of weather and climate models." Bulletin of the American Meteorological Society 98.3 (2017): 565-588.
   Franzke, Christian LE, et al. "Stochastic climate theory and modeling." Wiley Interdis- ciplinary Reviews: Climate Change 6.1 (2015): 63-78.

   **Answer:** We agree on this point, they have been added in the introduction.

15  ----

3. In the introduction the authors seem to distinguish between stochastic parameteri- zations and backscatter schemes. Backscatter schemes can also be stochastic so can be similar. I also think stochastic parameterizations are implicitly also based on the idea to "backscatter" energy from the unresolved scales into the resolved.

   **Answer:** Our point was not to distinguish the backscatter methods as being non-stochastic but that these method are

20  based on specific turbulence closure models which are different from the other non-specific method proposed. We have however restated in the text that we have stochastic version of these methods in mind:

   > They provide promising alternatives to other stochastic methods such as the ones based on the
   > reinjection of energy from the unresolved scale through backscatter schemes (Frederiksen and
   > Davies (1997); Frederiksen (1999), see also Frederiksen et al. (2017) for a recent review) or on
   > empirical stochastic modeling methods based on autoregressive processes (Arnold et al., 2013).

Of course, every stochastic methods can also be seen as a kind of backscatter scheme, since it introduces energy onto the large scales, coming in principle from the unresolved small-scales.
* * *
4. Page 2, footnote 2: I do not understand the meaning here.

**Answer:** This footnote just enumerate a few of the underlying hypothesis used to develop parameterization methods. For more clarity, we have incorporated this footnote in the text.
* * *
5. MAOOAM should be defined at first use.

**Answer:**

Ok.
* * *
6. Are "weak coupling" and "time scale separation" equivalent in a mathematical sense? How would one measure weak coupling in the real atmos-ocean system? For me weak coupling is rather opaque concept whereas time-scale separation is more tangible (at least I know how to estimate this from real data). Some comments on this would be appreciated.

**Answer:**

This is a very interesting question. While in some setting, a kind of equivalency can be made between time-scale separation and weak coupling (see Wouters et al. (2016)). On more general ground, such equivalence is far from trivial, and research could certainly be done in this direction.

Weak coupling is difficult to measure in data, and should rather be assessed by a proper modeling of the components of a system (dimensionalization). We agree that time-scale separation is easier to assess by computing the decorrelation time in data. We now comment at the end of Section 3.2 in the paper about that:

> *In some particular cases, it is possible to establish an equivalence between these two assumptions (Wouters et al., 2016). However, in general, the relation between the two is far from trivial. Time-scale separation is easy to assess, by considering the decorrelation times in the output data of the model. On the other hand, weak-coupling is difficult to measure in data, and appears in general as a small coupling parameter resulting from the proper modeling of the components of a system.*
* * *
7. In the "seamless MTV" procedure of Franzke et al. (2005) we generalized MTV and do not need to assume an Ornstein-Uhlenbeck process any longer but just one stochastic process with Gaussian statistics.

**Answer:** Once the formula have been obtained with an Ornstein-Uhlenbeck process, they can easily be reused in seamless way, considering Gaussian statistic. This is the kind of approach that have been taken in the article, considering the odd-moments of the $Y$-dynamics to be zero. We comment on Appendix Section A2 about this:

> *The formulas presented here are valid for an Ornstein-Uhlenbeck process, but as stated above, the covariance and correlations of the dynamics (A2) can be used directly as well, provided that the right assumptions are fulfilled. In the present work, we have always used the statistic of Eq. (A2) to compute the tensors.*

8. In Eqs. 36-39 are the coupling and time-scale separation parameters included. Which values have been used for the experiments?

    **Answer:** Yes, and as stated on page 10 line 22-25:

    > *In addition, the parameters $\delta$ and $\varepsilon$ appearing in Eqs. (36)-(39) will be set to 1, meaning that we consider the natural timescale separations and coupling strengths of the model. Nevertheless, the study of the impact of these parameters is important (Demaeyer and Vannitsem, 2018) and should be carried out in forthcoming works.*

10

9. (and 10.) In my work I use a split integration scheme: Runge-Kutta 4th order for the deterministic part and Euler-Maruyama for the stochastic part. Such a split scheme might solve parts of your numerical problems.

    In MTV the cubic terms are nonlinear damping. This has been shown in Majda, Andrew J., Christian Franzke, and Daan Crommelin. "Normal forms for reduced stochastic climate models." Proceedings of the National Academy of Sciences 106.10 (2009): 3649-3653. Peavoy, Daniel, Christian LE Franzke, and Gareth O. Roberts. "Systematic physics constrained parameter estimation of stochastic differential equations." Computational Statistics & Data Analysis 83 (2015): 182-199. Perhaps one can also ensure that the cubic term in WL is negative definite then the system should be stable.

20  **Answer:**

    Since the instability takes place in the deterministic integral part of the integro-differential equation, in the cubic terms, we do not think that it would solve the problem. In fact, with the same configuration and the same cubic terms, the MTV method does not diverge, showing that it is the nature of the WL equation which is a problem (integro-differential equation).

25  In addition, we think that the appelation "cubic terms" is more directly identifiable by the reader, but we mention now in the text:

*These cubic terms are nonlinear dampings, as shown in Majda et al. (2009) and Peavoy et al. (2015).*

We thank again the reviewer for his/her suggestions.

**References**

Arnold, H., Moroz, I., and Palmer, T.: Stochastic parametrizations and model uncertainty in the Lorenz'96 system, Philosophical Transactions of the Royal Society of London A: Mathematical, Physical and Engineering Sciences, 371, 20110 479, 2013.

Demaeyer, J. and Vannitsem, S.: Stochastic Parameterization of Subgrid-Scale Processes: A Review of Recent Physically Based Approaches, in: Advances in Nonlinear Geosciences, pp. 55–85, Springer, 2018.

Frederiksen, J., Kitsios, V., Okane, T., and Zidikheri, M.: Stochastic subgrid modelling for geophysical and three-dimensional turbulence, in: Nonlinear and Stochastic Climate Dynamics, edited by Franzke, C. and O'Kane, T., pp. 241–275, Cambridge University Press, 2017.

Frederiksen, J. S.: Subgrid-scale parameterizations of eddy-topographic force, eddy viscosity, and stochastic backscatter for flow over topography, Journal of the atmospheric sciences, 56, 1481–1494, 1999.

Frederiksen, J. S. and Davies, A. G.: Eddy viscosity and stochastic backscatter parameterizations on the sphere for atmospheric circulation models, Journal of the atmospheric sciences, 54, 2475–2492, 1997.

Majda, A. J., Franzke, C., and Crommelin, D.: Normal forms for reduced stochastic climate models, Proceedings of the National Academy of Sciences, 106, 3649–3653, 2009.

Peavoy, D., Franzke, C. L., and Roberts, G. O.: Systematic physics constrained parameter estimation of stochastic differential equations, Computational Statistics & Data Analysis, 83, 182–199, 2015.

Wouters, J., Dolaptchiev, S. I., Lucarini, V., and Achatz, U.: Parameterization of stochastic multiscale triads, Nonlinear Processes in Geophysics, 23, 435–445, 2016.

---

## Author Comment (AC3) · 18 Jun 2018

**Comparison of stochastic parameterizations in the framework of a coupled ocean-atmosphere model – Response to the 3rd reviewer**

Jonathan Demaeyer [1] and Stéphane Vannitsem [1]

[1]Institut Royal Météorologique de Belgique, Avenue Circulaire, 3, 1180 Brussels, Belgium

We thank the reviewer for his/her comment. We have taken the liberty of breaking down the few points he/she adresses as follow:

**1.1**  At the beginning of Section 3.1.2 the authors state that "These methods are applicable for parameterization purposes if the problem can be cast into a backward Kolmogorov equation." This is not a correct statement: Every dynamical system (deterministic or stochastic) can be cast into a backward Kolmogorov equation. Casting a nonlinear dynamical system into the linear backward Kolmogorov equation (or alternatively the linear Fokker-Planck equation) allows for the machinery of perturbation theory of linear systems to derive explicit formulae for the homogenized equations. The authors may want to reference here the monograph "Multiscale Methods: Averaging and Homogenization" by Pavliotis and Stuart.

**Answer:** We agree with the referee and have modified the statement as:

> *These methods are applicable for parameterization purposes if the problem can be cast into a linear backward Kolmogorov equation (Pavliotis and Stuart, 2008).*
* * *
**1.2** Furthermore, the authors provide references for the mathematical justification of MTV (including Papanicolaou). These are references if the underlying dynamical system is stochastic; for the applicability of the MTV procedure for deterministic systems, the references are Melbourne and Stuart (2011), Nonlinearity 24, pp 1361, Gottwald and Melbourne (2013), Proc. Roy. Soc. A 469, pp 2013020 and Kelly and Melbourne (2017), Journal of Functional Analysis 10, pp 4063.

**Answer:** We agree and have added the references:

> *This approach is based on the singular perturbation methods that were developed for the analysis of the linear Boltzmann equation in an asymptotic limit (Grad, 1969; Ellis and Pinsky, 1975; Papanicolaou, 1976; Majda et al., 2001) and it has been applied to deterministic systems as well (Melbourne and Stuart, 2011; Kelly and Melbourne, 2017).*
* * *
**1.3** The choice of notation with $\rho$ for the backward Kolmogorov equation which propagates expectation values, is odd. $\rho$ is usually reserved for densities. I also found the backward in time integration with the expectation value being defined at

the final time cumbersome; why not have a positive sign on the left hand side of (25) and (26), and have the expectation value equal to $f(X)$ at $t = 0$? Also, the generator $\mathcal{L}$ is not defined when it is introduced.

**Answer:** Please see our answer to the 1st referee. For the choice of the backward integration, we align with the MTV derivation in Majda et al. (2001). We have modified the first sentence where the operators appear as:

*In this setting, the backward Kolmogorov equation reads (Majda et al., 2001):*

$$-\frac{\partial \rho^\delta}{\partial s} = \left[ \frac{1}{\delta^2} \mathcal{L}_1 + \frac{1}{\delta} \mathcal{L}_2 + \mathcal{L}_3 \right] \rho^\delta$$

*where the probability density $\rho^\delta(s, \boldsymbol{X}, \boldsymbol{Y}|t)$ is defined with the final value problem $f(\boldsymbol{X})$:*
*$\rho^\delta(t, \boldsymbol{X}, \boldsymbol{Y}|t) = f(\boldsymbol{X})$. The density $\rho^\delta$ can be expanded in term of $\delta$ and inserted in Eq. (25). The zeroth order of this equation $\rho^0$ can be shown to be independent of $\boldsymbol{Y}$ and its evolution given by a closed, averaged backward Kolmogorov equation (Kurtz, 1973):*

$$-\frac{\partial \rho^0}{\partial s} = \bar{\mathcal{L}} \rho^0 \,.$$

*The precise definition of the operators $\mathcal{L}_i$ and $\bar{\mathcal{L}}$ acting on the densities is given in Appendix A.*
* * *
2. In the case when the unresolved scales consist of the wavenumber 2 atmospheric variables, the authors found that the WL approach leads to unstable dynamics. I am surprised by that. The authors relate this to a cubic form. Can the authors comment on which assumption of WL is being violated by this choice of unresolved variables? Are higher-order corrections needed?

   **Answer:**

   The weak-coupling assumption is violated in the atmosphere when $\varepsilon$ is set to 1 as we have done for every experiment in the article. This might be a reason and a simple way (but computationnaly expensive) to check this is to repeat the experiment with smaller values of $\varepsilon$ until the instability disappears. Higher-order corrections might solve the problem but are quite tedious to compute. What we find the most intriguing is the fact that the MTV method is stable, but not the WL one, leading us to suspect that the method is sensitive to the accuracy of the memory term $M_3$ integration.

3. There are a few typos in the text:
   Eqn (24): missing full stop.
   Above (42): let -> left

   **Answer:**

   Thanks.

We thank again the reviewer for his/her suggestions.

**References**

Ellis, R. S. and Pinsky, M. A.: The first and second fluid approximations to the linearized Boltzmann equation, J. Math. Pures Appl, 54, 125–156, 1975.

Grad, H.: Singular and nonuniform limits of solutions of the Boltzmann equation, Transport theory, 1, 269–308, 1969.

5 Kelly, D. and Melbourne, I.: Deterministic homogenization for fast–slow systems with chaotic noise, Journal of Functional Analysis, 272, 4063–4102, 2017.

Kurtz, T. G.: A limit theorem for perturbed operator semigroups with applications to random evolutions, Journal of Functional Analysis, 12, 55–67, 1973.

Majda, A. J., Timofeyev, I., and Vanden Eijnden, E.: A mathematical framework for stochastic climate models, Communications on Pure and 10 Applied Mathematics, 54, 891–974, 2001.

Melbourne, I. and Stuart, A.: A note on diffusion limits of chaotic skew-product flows, Nonlinearity, 24, 1361, 2011.

Papanicolaou, G. C.: Some probabilistic problems and methods in singular perturbations, Journal of Mathematics, 6, 1976.

Pavliotis, G. and Stuart, A.: Multiscale methods: averaging and homogenization, Springer Science & Business Media, 2008.